# Modeling spinal locomotor circuits for movements in developing zebrafish

Yann Roussel[1,2], Stephanie F Gaudreau[1], Emily R Kacer[1], Mohini Sengupta[3], Tuan V Bui[1]*

[1]Brain and Mind Research Institute, Centre for Neural Dynamics, Department of Biology, University of Ottawa, Ottawa, Canada; [2]Blue Brain Project, École Polytechnique Fédérale de Lausanne, Genève, Switzerland; [3]Washington University School of Medicine, Department of Neuroscience, St Louis, United States

**Abstract** Many spinal circuits dedicated to locomotor control have been identified in the developing zebrafish. How these circuits operate together to generate the various swimming movements during development remains to be clarified. In this study, we iteratively built models of developing zebrafish spinal circuits coupled to simplified musculoskeletal models that reproduce coiling and swimming movements. The neurons of the models were based upon morphologically or genetically identified populations in the developing zebrafish spinal cord. We simulated intact spinal circuits as well as circuits with silenced neurons or altered synaptic transmission to better understand the role of specific spinal neurons. Analysis of firing patterns and phase relationships helped to identify possible mechanisms underlying the locomotor movements of developing zebrafish. Notably, our simulations demonstrated how the site and the operation of rhythm generation could transition between coiling and swimming. The simulations also underlined the importance of contralateral excitation to multiple tail beats. They allowed us to estimate the sensitivity of spinal locomotor networks to motor command amplitude, synaptic weights, length of ascending and descending axons, and firing behavior. These models will serve as valuable tools to test and further understand the operation of spinal circuits for locomotion.

*For correspondence: tuan.bui@uottawa.ca

Competing interests: The authors declare that no competing interests exist.

## Introduction

Movements made in the early stages of development can be critical for the survival of many species. The escape response seen in various fish and amphibians is one such example of a vital movement present at early developmental stages (*Domenici and Hale, 2019*). However, the nervous system's control of movement does not come fully formed but matures as the nervous system develops (*Favero et al., 2014*). This maturation enables a broader repertoire of movements to arise. During this process, new neurons are born and subsequently integrated into neural circuits that are newly formed or refined, presumably leading to the emergence of progressively more coordinated and skillful maneuvers. Determining how the assembly of new circuits leads to the emergence of new movements can provide valuable insights into the role of distinct neurons or circuits in motor control.

The maturation of swimming in developing zebrafish has been well described at both the etho-logical and the cellular levels (*Drapeau et al., 2002*; *McLean and Fetcho, 2009*). Single strong body bends on one side of the body, also known as coils, emerge during the first day of development at around 17 hr post-fertilization (hpf) as the earliest locomotor behavior (*Saint-Amant and Drapeau, 1998*). Single coils are quickly followed by double coils (i.e., two successive coils, one for each side of the body) at around 24 hpf (*Knogler et al., 2014*). Touch-evoked swimming appears around 27 hpf as coiling begins to subside. Spontaneous swimming movements emerge around 2–3 days post-fertilization (dpf) (*Saint-Amant, 2010*). The first swimming movement zebrafish exhibit is burst

**eLife digest** The spinal cord is a column of nerve tissue that connects the brain to the rest of the body in vertebrate animals. Nerve cells in the spinal cord, called neurons, help to control and coordinate the body's movements. As the spinal cord develops, new neurons are born and new connections are made between neurons and muscles, resulting in more coordinated and skillful movements as time goes on.

Zebrafish, for example, display body-bending maneuvers called coils within 24 hours of the egg being fertilized. Next, bursts of swimming movements emerge, which are driven by sporadic tail beats. These tail maneuvers become more consistent as the fish develops, and eventually result in smooth movements called beat-and-glide swimming. The groups of spinal cord neurons that appear at each stage of zebrafish development have been characterized, but it remains unclear how newly formed circuits (groups of neurons recently connected to each other) work together to produce swimming maneuvers.

To answer this question, Roussel et al. simulated changes in the spinal cord that help zebrafish acquire new swimming movements as they grow. The computer models encoded neural circuits based on cell populations identified in experimental studies, and replicated swimming behaviors that emerge during the first few days of zebrafish development. Simulations tested how specific neural circuits generate the characteristic swimming movements that represent key developmental milestones in zebrafish.

The results showed that adding new neurons and more cell-to-cell connections led to increasingly sophisticated swimming maneuvers. As the zebrafish spinal cord matured, the fish were better able to control the pace and duration of their swimming movements. Roussel et al. also identified specific patterns of neural activity linked to particular maneuvers. For example, tail beats switch direction when neurons on one side of the spinal cord excite neurons on the opposite side. This activity, which becomes more rhythmic, also needs to be exquisitely timed to produce and coordinate the right motion.

Roussel et al.'s modelling of developmental milestones in growing zebrafish provides insights into how neural networks control movement. The computer models are among the first to accurately reproduce swimming behaviors in developing zebrafish. More experimental data could be added to the models to capture the full range of early zebrafish movements, and to further investigate how maturing spinal cord circuits control swimming. Since zebrafish and mammals have many spinal neurons in common, further research may aid our understanding of movement disorders in humans.

swimming characterized by long (1 s long) but infrequent episodes of tail beats. Burst swimming is then replaced by beat-and-glide swimming characterized by shorter (several hundreds of milliseconds long) but more frequent episodes. In both cases, swim episodes consist of repetitive left-right alternating, low-amplitude tail beats that propagate from the rostral toward the caudal end of the fish body and are generated at 20 to 80 Hz (**Budick and O'Malley, 2000**; **Buss and Drapeau, 2001**).

During this rapid series of transitions between locomotor maneuvers, populations of spinal neurons are progressively generated, starting with primary motoneurons (MNs) at about 9 hpf. Subsequently, spinal MNs and interneurons (INs) are generated in stereotyped spatiotemporal birth orders (**Kimmel et al., 1994**; **Myers et al., 1986**; **Satou et al., 2012**). Two successive waves of axogenesis occur in the embryonic spinal cord (**Bernhardt et al., 1990**). The first wave occurs around 16–17 hpf. It includes axon growth in primary MNs that innervate red and white muscle fibers at early developmental stages (**Buss and Drapeau, 2000**). Primary MNs enable coiling and escape movements (**Kimmel et al., 1995**; **Saint-Amant and Drapeau, 2000**). Several spinal INs that are also important for early movements extend their axons along with primary MNs. These include Ipsilateral Caudal (IC) INs that are thought to play an essential role in driving the rhythm of early locomotor behavior due to their endogenous bursting activity (**Tong and McDearmid, 2012**). The second wave of axon growth occurs at around 23–25 hpf. It involves axon growth in secondary MNs involved with slower movements (**Liu and Westerfield, 1988**) and spinal IN populations that include excitatory and inhibitory, ipsilaterally and contralaterally, and ascending and descending projecting subtypes (**Bernhardt et al., 1990**; **Higashijima et al., 2004**). The progressive generation of new populations

of spinal neurons and continued axonal growth coincides with the expansion of the zebrafish loco-motor repertoire. This timing suggests that incorporating spinal circuits into existing locomotor cir-cuits underlies the acquisition of novel locomotor maneuvers.

We have recently provided evidence that the maturation from coiling to later stages of swimming is accompanied by an operational switch in how spinal locomotor circuits generate the rhythm underlying tail beats. Specifically, we demonstrated that spinal circuits transitioned from relying upon pacemakers with endogenous bursting properties during coiling toward depending upon net-work oscillators whose rhythm is driven by excitatory and inhibitory synapses (*Roussel et al., 2020*). In light of these and earlier findings describing the composition and maturation of spinal locomotor circuits, we sought to generate computational models that replicate developmental locomotor movements of the zebrafish. We iteratively constructed models for several locomotor movements by incorporating specific spinal populations, shifts in relative connection strength, and changes in the firing behavior of neurons. While computational modeling has generated invaluable insights into the function and mechanisms of spinal locomotor circuits of several species (*Ausborn et al., 2019*; *Bicanski et al., 2013*; *Danner et al., 2019*; *Ferrario et al., 2018*; *Hull et al., 2016*; *Kozlov et al., 2014*; *Sautois et al., 2007*), there is to our knowledge no such model for the developing zebrafish spinal cord. Here, we build some of the first computational models of the zebrafish spinal locomotor circuit that can accurately reproduce predominant locomotor behaviors during early zebrafish devel-opment. In the process, we test theories about the possible contributions of specific neural circuits and spinal populations to locomotor movements in zebrafish and identify untested hypotheses on the operation of spinal locomotor networks in developing zebrafish.

## Results

We aimed to model how new locomotor movements may emerge from the integration of spinal INs and the modification of synaptic weights and firing behavior over the first few days of development in the zebrafish. Our approach was to build an initial model based upon previously reported experi-mental observations of spinal circuits when the first locomotor movements emerge in zebrafish around 1 dpf. We then successively built upon this initial model to replicate several locomotor maneuvers of the developing zebrafish.

The models were composed of single-compartment neurons whose firing dynamics were deter-mined by a small set of differential equations (*Izhikevich, 2007*). The firing of MNs was converted to muscle output. This output was used to estimate body angle and locomotor activity during simula-tions (*Figure 1*). The composition of each model depended on the developmental stage and the locomotor movement to be generated.

### Single coiling (>17 hpf) results from unilateral gap junction coupling

Coiling, which is already observed at 1 dpf, is characterized by a single strong, slow (hundreds of milliseconds in duration) tail beat on one side of the body followed by a return to resting position (*Saint-Amant and Drapeau, 1998*). Coiling events are relatively infrequent, reaching a maximum fre-quency of 1 Hz around 20 hpf (*Saint-Amant and Drapeau, 1998*). Previous studies have established that this behavior is generated by a spinal circuit relying primarily on gap junctions (i.e., electrical synapses) (*Saint-Amant and Drapeau, 2001*). It has been proposed that rostrally located IC pace-maker spinal neurons (*Tong and McDearmid, 2012*) drive periodic depolarizations (PDs) of ipsilat-eral MNs via electrical synapses (*Drapeau et al., 2002*; *Saint-Amant and Drapeau, 2001*). Glycinergic synaptic bursts (SBs) are observed in MNs during contralateral coiling events (*Saint-Amant and Drapeau, 2001*). These SBs have been proposed to arise from contralaterally projecting glycinergic neurons (*Saint-Amant and Drapeau, 2000*) but are not responsible for any action poten-tial firings or coiling movements (*Saint-Amant and Drapeau, 2001*). Applying a gap junction blocker, heptanol, but not glutamatergic and glycinergic antagonists, suppressed spinal activity responsible for coiling (*Saint-Amant and Drapeau, 2000*).

### Network description

(*Figure 2A*) Based on the experimental observations reported above, the model for single coiling consisted of rostrocaudal chains of electrically coupled spinal neurons driven by a kernel of five recurrently connected pacemakers (IC neurons). One chain consisted of 10 MNs. The other chain

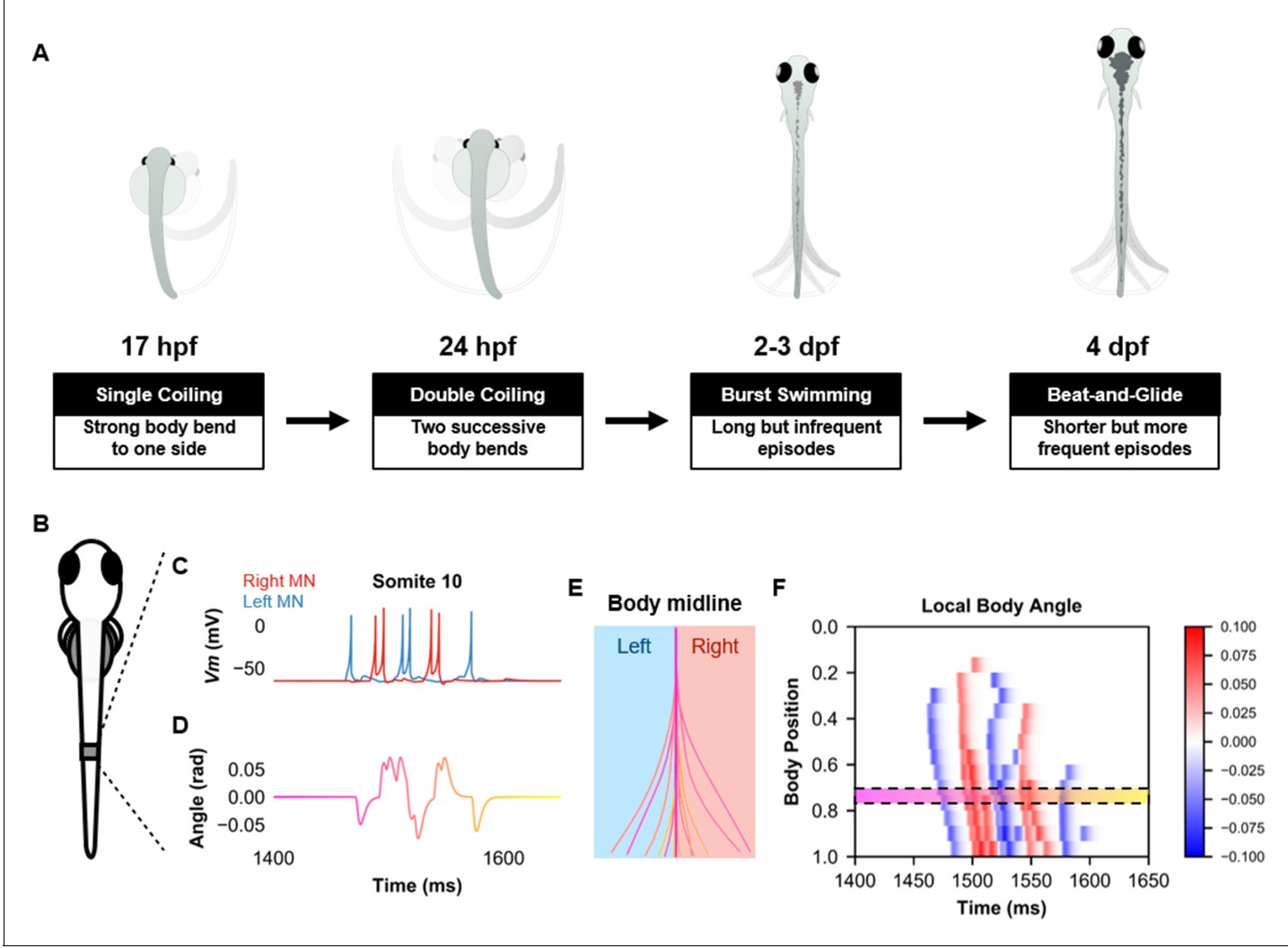

**Figure 1.** Simulation of the spinal locomotor circuit coupled to a musculoskeletal model during a beat-and-glide swimming episode. (**A**) Schematic of locomotor movements during the development of zebrafish. (**B**) Schematic of a fish body with 10th somite outlined. (**C**) Motoneuron membrane potential ($Vm$) in the 10th somite during a single beat-and-glide swimming episode from our model is used to calculate this body segment's body angle variation (**D**) in a musculoskeletal model. (**E**) Several representative body midlines from this episode of beat-and-glide swimming. Body midline is computed by compiling all the calculated local body angles along the simulated fish body. (**F**) Heat-map of local body angle (in radians) across the total body length and through time during the episode. Red is for right curvatures, while blue labels left curvatures. Body position on the ordinate, 0 is the rostral extremity, while 1 is the caudal extremity. In (**D–F**), the magenta to yellow color coding represents the progression through the swimming episode depicted.

consisted of 10 contralaterally projecting commissural inhibitory neurons. Neurons from the V0d population are active during large amplitude movements such as escapes (*Satou et al., 2020*), and so we assumed V0ds were the commissural inhibitory neurons active during coiling, which is another large amplitude movement. We selected an IC kernel size of five as a trade-off between computational simplicity and robustness of the kernel to the failure of firing of a small number of cells. Similarly, the size of the coiling model was set to 10 somites. Thus, each model somite represents approximately three biological somites. This choice was made as a trade-off between computational simplicity and recreating the kinematics of coiling fish.

IC neurons have been reported to project caudally through multiple somites (*Bernhardt et al., 1990*). Therefore, in addition to their recurrent connections, each IC formed electrical synapses with several rostral MNs and V0ds (the first four of each ipsilateral chain in our model). Electrical coupling between many populations of early born spinal neurons has been previously demonstrated, including between IC and MNs (*Saint-Amant and Drapeau, 1998*). Coupling between IC neurons and

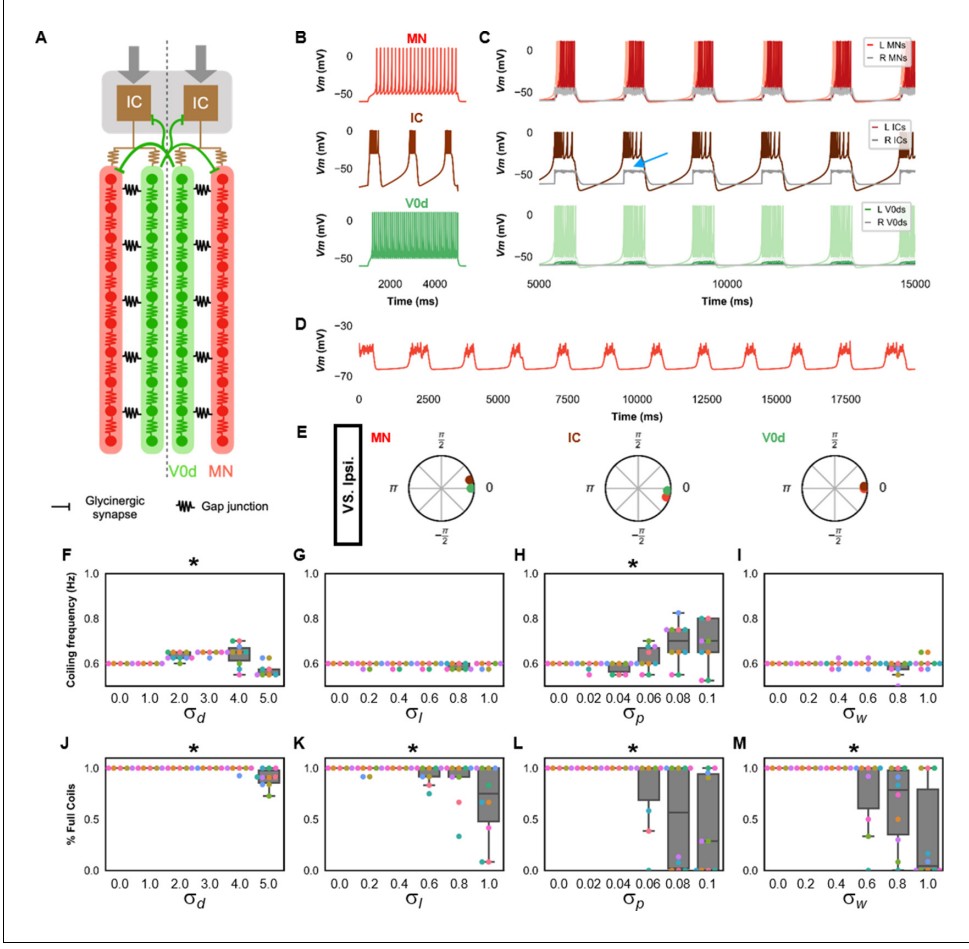

**Figure 2.** Single coiling model driven by pacemaker neurons. (**A**) Schematic of the single coiling model. The dashed line indicates the body midline. Gray arrows indicate descending motor command. (**B**) Membrane potential (*Vm*) response of isolated spinal neuron models to a depolarizing current step. (**C**) *Vm* of spinal neurons during a simulation with a tonic command to left pacemakers only. Note the synaptic bursts in gray in the right MNs and IC neurons (a blue arrow marks an example). The *Vm* of a rostral (lightest), middle, and caudal (darkest) neuron is shown, except for IC neurons that are all in a rostral kernel. (**D**) Periodic depolarizations in a hyperpolarized motoneuron on the same side where single coils are generated. (**E**) The phase delay of left neurons in relation to ipsilateral spinal neurons in the first somite and an IC in the rostral kernel in a 10,000 ms simulation. The reference neuron for each polar plot is labeled, and all neurons follow the same color-coding as the rest of the figure. A negative phase delay indicates that the reference neuron precedes the neuron to which it is compared. A phase of 0 indicates that a pair of neurons is in-phase; a phase of π indicates that a pair of neurons is out-of-phase. Sensitivity testing showing (**F–I**) coiling frequency and (**J–M**) proportion of full coils during ten 20,000 ms simulation runs at each value of $\sigma_d$, $\sigma_l$, $\sigma_p$, and $\sigma_w$ tested. Each run is color-coded. Bars on box plots represent 25th, median, and 75th percentile. Whiskers extend to 1.5 times the interquartile range. L: left, R: right. *Statistics*: Asterisks denote significant differences detected using a one-factor ANOVA test. (**F**) $F_{5,59}$=10.4, $p$=5.2×10$^{-7}$. (**G**) $F_{5,59}$=2.4, $p$=0.05. (**H**) $F_{5,59}$=5.2, $p$=0.0006. (**I**) $F_{5,59}$=2.2, $p$=0.07. (**J**) $F_{5,59}$=10.9, $p$=2.7×10$^{-7}$. (**K**) $F_{5,59}$=4.9, $p$=0.0009. (Note that there were no pairwise differences detected). (**L**) $F_{5,59}$=6.5, $p$=8.2×10$^{-5}$. (**M**) $F_{5,59}$=8.8, $p$=3.5×10$^{-6}$. *P*-values for *t*-tests are found in *Figure 2—source data 1*. See also *Figure 2—figure supplements 1* and *2* and *Figure 2—videos 1* and *2*. IC, Ipsilateral Caudal; MN, motoneuron.

The online version of this article includes the following video, source data, and figure supplement(s) for figure 2:

**Source data 1.** P-values for sensitivity testing in single coiling model.
**Figure supplement 1.** Silencing spinal neurons during single coiling.
**Figure supplement 2.** Membrane potential (*Vm*) during a simulation of a 30-somite single coiling model.
**Figure 2—video 1.** Single coiling model.
https://elifesciences.org/articles/67453#fig2video1
**Figure 2—video 2.** Truncated coils.
*Figure 2 continued on next page*

commissural inhibitory neurons has not been demonstrated yet. We based this electrical coupling between ICs and V0ds on the fact that glutamatergic blockers do not block glycinergic SBs present at this stage (*Saint-Amant and Drapeau, 2001*), suggesting that gap junctions mediate the activation of V0ds underlying these glycinergic bursts. Gap junction weights are found in *Table 1* .

The connectivity within the chains was identical for both MN and V0d chains. Each neuron in a chain formed electrical synapses with its three nearest rostral and caudal neighbors within the same chain. There was also electrical coupling across the two ipsilateral chains as MNs formed gap junctions with the three nearest rostral and three nearest caudal V0ds and vice-versa. Paired recordings of MNs and V0ds at this stage have yet to be published. Our assumption that MNs and V0ds are electrically coupled at this stage was based upon the widespread electrical coupling between ipsilateral spinal neurons (*Saint-Amant and Drapeau, 2001*). To reproduce the glycinergic bursts observed in MNs at this stage (*Saint-Amant and Drapeau, 2001*), V0ds projected to contralateral MNs. Thus, V0ds formed glycinergic synapses with contralateral MNs and ICs. The reversal potential of glycinergic synapses ($E_{gly}$) is depolarized during development (*Ben-Ari, 2002*) and was set to −45 mV in the single coiling model (see *Table 2*). All V0ds sent ascending projections to contralateral ICs. V0ds projected to contralateral MNs within five to six segments so that the $i$th V0d projected to all contralateral MNs between the $i−5$ and $i+5$ segments. Chemical synaptic weights are found in *Table 3*.

Each neuron was modeled as a single compartment neuron with subthreshold and suprathreshold membrane potential dynamics described by a small set of differential equations (*Izhikevich, 2007*). These equations have nine parameters: *a*, *b*, *c*, *d*, and $V_{max}$ (which respectively represent the time scale of the recovery variable *u*, the sensitivity of *u* to the subthreshold variation of *V*, the reset value of *V* after a spike, the reset value of *u*, and the action potential peak), and *k*, *C*, $V_r$, and $V_t$ (coefficient for the approximation of the subthreshold part of the fast component of the current-voltage relationship of the neuron, cell capacitance, resting membrane potential, and threshold of action potential firing). Parameter values of ICs (see *Table 4* for all neuron parameters) were chosen such that they exhibited a relatively depolarized threshold of action potentials and bursts of short action potentials lasting hundreds of milliseconds as seen in experimental recordings in embryonic zebrafish (*Tong and McDearmid, 2012*). They were also modeled to exhibit periodic bursts lasting hundreds of milliseconds in response to a constant tonic drive (*Figure 2B*). This firing pattern was generated in part by having a low value of *a* and a relatively depolarized value of *c*. MNs (*Drapeau et al., 1999*) and V0ds were modeled to generate tonic repetitive firing in response to a

**Table 1.** Electrical synapse (gap junctions) weights between neuron populations.

| Coiling<br>Beat-and-glide | MN | IC | V0d dI6 | V0v | V2a |
|---|---|---|---|---|---|
| **MN** | | | | | |
| Single coiling | 0.1 | | | | |
| Double coiling | 0.07 | | | | |
| Beat-and-glide (all models) | 0.005 | | | | |
| **IC** | | | | | |
| Single coiling | 0.04 | 0.001 | | | |
| Double coiling | 0.03 | 0.0001 | | | |
| **V0d or dI6** | | | | | |
| Single coiling | 0.01 | 0.05 | 0.04 | | |
| Double coiling | 0.0001 | 0.05 | 0.04 | | |
| Beat-and-glide (all models) | 0.0001 | | 0.04 | | |
| **V0v** | | | | | |
| Double coiling | 0.0001 | 0.0005 | | 0.05 | |
| Beat-and-glide (all models) | 0.005 | | | 0.05 | |
| **V2a** | | | | | |
| Double coiling | 0.005 | 0.15 | | | 0.005 |
| Beat-and-glide (all models) | 0.005 | | | | 0.005 |

**Table 2.** Glutamatergic and glycinergic reversal potentials and time constants.

| Chemical synapse | $E_{rev}$ | $\tau_r$ | $\tau_f$ | $V_{thr}$ |
|---|---|---|---|---|
| Glutamatergic | 0 | 0.5 | 1.0 | −15 |
| Glycinergic | −45, −58, −70[*] | 0.5 | 1.0 | −15 |

[*]For single and double coiling and beat-and-glide swimming models, respectively.

step depolarization (*Figure 2B*). Finally, to activate the circuit, a constant drive was provided to the left ICs only. Restricting the drive to left ICs prevented the appearance of near-coincident bilateral coils that could be misinterpreted as spinally mediated multiple coils.

## Simulations results

Our simulations show that this model can generate single coils characterized by large body bends to one side of the body lasting approximately 1 s (*Figure 2C*, *Figure 2—video 1*). Our base single coiling model generated six evenly interspersed single coils during a 10 s simulation. This 0.6 Hz coiling frequency is within the 0–1.0 Hz range of frequencies observed during zebrafish development (*Saint-Amant and Drapeau, 1998*; *Saint-Amant and Drapeau, 2000*). Silencing ICs blocked activity in all spinal neurons (*Figure 2—figure supplement 1A*), emphasizing the central role of the IC kernel in the generation of single coils.

Previously reported whole-cell patch-clamp recordings of MNs at this developmental stage display two types of events (*Saint-Amant and Drapeau, 2000*; *Saint-Amant and Drapeau, 2001*): PDs via electrical synapses and SBs from contralateral spinal glycinergic neurons that are depolarizing at rest due to the depolarized chloride reversal potential observed early in development. These events last hundreds of milliseconds. In our model, SBs were observed in the contralateral ICs and MNs (events during coilings in left neurons seen in the gray traces in *Figure 2C*). SBs were caused by

**Table 3.** Chemical synapse weights between neuron populations.
Pre-synaptic neurons are in rows. Post-synaptic neurons in columns.

| | Post-synaptic | | | | | | |
|---|---|---|---|---|---|---|---|
| Pre-synaptic | MN | IC | V0d dl6 | V0v | V2a | V1 | Muscle |
| **MN** | | | | | | | |
| Single coiling | | | | | | | 0.015 |
| Double coiling | | | | | | | 0.02 |
| Beat-and-glide (all models) | | | | | | | 0.1 |
| **V0d** | | | | | | | |
| Single coiling | 0.3 | 0.3 | | | | | |
| Double coiling | 2.0 | 2.0 | | | 2.0 | | |
| **dl6** | | | | | | | |
| Beat-and-glide (base) | 1.5 | | 0.25* | | 1.5 | | |
| Beat-and-glide (bursting V2a) | 1.5 | | 0.25* | | 2.0 | | |
| Beat-and-glide (all tonic neurons) | 1.5 | | 0.25* | | 1.5 | | |
| **V0v** | | | | | | | |
| Double coiling | | 0.175 | | | | | |
| Beat-and-glide (base) | | | | | 0.4 | | |
| Beat-and-glide (bursting V2a) | | | | | 0.75 | | |
| Beat-and-glide (all tonic neurons) | | | | | 0.4 | | |
| **V2a** | | | | | | | |
| Double coiling | | | | 0.04 | | | |
| Beat-and-glide (base) | 0.5 | | 0.5 | 0.3 | 0.3 | 0.5 | |
| Beat-and-glide (bursting V2a) | 0.5 | | 0.75 | 0.275 | 0.3 | 0.5 | |
| Beat-and-glide (all tonic neurons) | 0.5 | | 0.5 | 0.25 | 0.3 | 0.5 | |
| **V1** | | | | | | | |
| Beat-and-glide (base) | 1.0 | | 0.2 | 0.1 | 0.5 | | |
| Beat-and-glide (bursting V2a) | 1.0 | | 0.2 | 0.1 | 0.5 | | |
| Beat-and-glide (all tonic neurons) | 1.0 | | 0.2 | 0.1 | 0.6 | | |

*Scaled by a random number selected from a gaussian distribution with mean of 1 and variance of 0.1.

**Table 4.** Parameter values of neurons.

| Model population | a | b | c | d | $V_{max}$ | $V_r$ | $V_t$ | k | C | Rostro-caudal position[*] | $I_{drive}$[†] |
|---|---|---|---|---|---|---|---|---|---|---|---|
| MN | | | | | | | | | | 5.0+1.6*n | |
| Single coiling | 0.5 | 0.1 | −50 | 0.2 | 10 | −60 | −45 | 0.05 | 20 | | |
| Double coiling | 0.5 | 0.1 | −50 | 100 | 10 | −60 | −50 | 0.05 | 20 | | |
| Beat-and-glide | 0.5 | 0.01 | −55 | 100 | 10 | −65 | −58 | 0.5 | 20 | | |
| IC | | | | | | | | | | 1.0 | |
| Single coiling | 0.0005 | 0.5 | −30 | 5 | 0 | −60 | −45 | 0.05 | 50 | | 50 |
| Double coiling | 0.0002 | 0.5 | −40 | 5 | 0 | −60 | −45 | 0.03 | 50 | | 35 |
| V0d | | | | | | | | | | 5.0+1.6*n | |
| Single coiling | 0.5 | 0.01 | −50 | 0.2 | 10 | −60 | −45 | 0.05 | 20 | | |
| Double coiling | 0.02 | 0.1 | −30 | 3.75 | 10 | −60 | −45 | 0.09 | 6 | | |
| dI6 | | | | | | | | | | 5.1+1.6*n | |
| Beat-and-glide (all models) | 0.1 | 0.002 | −55 | 4 | 10 | −60 | −54 | 0.3 | 10 | | |
| V0v | | | | | | | | | | 5.1+1.6*n | |
| Double coiling | 0.02 | 0.1 | −30 | 11.6 | 10 | −60 | −45 | 0.05 | 20 | | |
| Beat-and-glide (base) | 0.01 | 0.002 | −55 | 8 | 10 | −60 | −54 | 0.3 | 10 | | |
| Beat-and-glide (bursting V2a and all tonic models) | 0.1 | 0.002 | −55 | 4 | 10 | −60 | −54 | 0.3 | 10 | | |
| V2a | | | | | | | | | | 5.1+1.6*n | |
| Double coiling | 0.5 | 0.1 | −40 | 100 | 10 | −60 | −45 | 0.05 | 20 | | 2.89 |
| Beat-and-glide (base and all tonic models) | 0.1 | 0.002 | −55 | 4 | 10 | −60 | −54 | 0.3 | 10 | | 3.05 |
| Beat-and-glide (bursting V2a model) | 0.01 | 0.002 | −55 | 8 | 10 | −60 | −54 | 0.3 | 10 | | |
| V1 | | | | | | | | | | 7.1+1.6*n | |
| Beat-and-glide (all models) | 0.1 | 0.002 | −55 | 4 | 10 | −60 | −54 | 0.3 | 10 | | |

[*]n=0 to N−1, N being the total number of neurons in that given population.

[†]Amplitude of tonic motor command drive.

glycinergic input from V0ds activated during the ipsilateral coilings. As observed experimentally (*Saint-Amant and Drapeau, 2001*), preventing SBs by silencing glycinergic synapses from V0ds did not preclude the generation of single coiling, nor did it lead to the generation of multiple coilings (*Figure 2—figure supplement 1C*). PDs can be unmasked by hyperpolarizing MNs sufficiently to prevent the firing of action potentials (*Figure 2D*). An analysis of the phase delays between ipsilateral neurons during single coils shows that IC neuron firing precedes ipsilateral MN and V0d firing (*Figure 2E*) and reinforces that ICs drive single coiling events.

To further validate the model, we tested whether the model could still generate single coils with different parameters. First, we tested whether the model could still generate single coils when the number of model somites was increased from 10 to 30 to be closer to the number of biological somites in zebrafish (*Stickney et al., 2000*). A 30-somite model with IC axons extending to all somites and several modified gap junction weights (*Table 1*) generated single coils (*Figure 2—figure supplement 2*).

Next, the base model's sensitivity to within-model parameter variability was tested. Variability in the amplitude of the tonic motor command, the rostrocaudal extent of every axonal projection, every parameter that set the dynamics of the membrane potential of each neuron (*a*, *b*, *c*, *d*, and $V_{max}$, *k*, *C*, $V_r$, and $V_t$), and all of the weights of gap junction and chemical synapses were modeled by scaling each value by a random number picked for each simulation. The random numbers were derived from a Gaussian distribution with mean =1, and standard deviations, $\sigma_d$ (tonic drive), $\sigma_l$ (rostrocaudal length of axonal projections), $\sigma_p$ (dynamics of membrane potential), and $\sigma_w$ (synaptic weights), respectively. Ten 20-s long simulations were run at various values of $\sigma_d$, $\sigma_l$, $\sigma_p$, and $\sigma_w$. In each simulation, the variability of only one of the four sets of parameters (amplitude of motor drive, length of axonal projection, membrane potential dynamics, and synaptic weights) was tested, and the standard deviations of the three other sets of parameters were set to 0.

The single coiling model's suitability was assessed by the relative absence of truncated coils, which were movements with only partial contractions restricted to the body's rostral segments (*Figure 2—video 2*). We sought to determine the upper limit of variability within which the single coiling model remained suitable. For this reason, the ranges of $\sigma_d$, $\sigma_l$, $\sigma_p$, and $\sigma_w$ that were tested differed amongst the four sets of parameters tested (*Figure 2F-M*). A comparison of the level of variability at

which the models start generating more varying frequency of coiling and more truncated coils suggests that the single coiling model is more robust to noise in the amplitude of the tonic motor command (*Figure 2F, J*) and was most sensitive to variability in the parameters governing the dynamics of the membrane potential (*Figure 2H, L*). The single coiling model was relatively mildly sensitive to variability in the synaptic weights and the rostrocaudal extent of the axon projections (*Figure 2G, I, K, M*).

Overall, the model replicated this first locomotor behavior of zebrafish in terms of the duration and frequency of coiling events as well as synaptic events of MNs. We then built upon this model to replicate the next step in the development of locomotion: the appearance of double coiling.

## Double coiling (>24 hpf) depends on the timing and strength of contralateral excitation and inhibition

After single coils appear, double coils emerge as a transitory locomotor behavior at around 24 hpf, coexisting with the single coiling behavior (*Knogler et al., 2014*). Double coiling is characterized by two successive coils, one on each side of the body, and lasts about 1 s (*Knogler et al., 2014*). Eventually, double coiling becomes the predominant coiling behavior. Double coiling can represent nearly three-quarters of all coiling events at its peak frequency, with the rest mainly being single coils (*Knogler et al., 2014*).

At the stage when double coiling appears (24 dpf), the previous electrical scaffold for single coils seems to be supplemented with chemical glutamatergic synapses to form a hybrid electrical-chemical circuit (*Knogler et al., 2014*). Blocking glutamatergic transmission precludes double coils while sparing single coils (*Knogler et al., 2014*). In contrast, blocking glycinergic synapses led to triple or even quadruple coils (*Knogler et al., 2014*). These experimental observations suggest that synaptic excitation is required for successive coils after a first coil. Glycinergic transmission seems to prevent the generation of more than two successive coils. Patch-clamp recordings of MNs at this developmental stage exhibit the same isolated PDs and SBs from earlier developmental stages and show mixed events in which a PD event immediately follows an SB or vice-versa (*Knogler et al., 2014*). Interestingly, the application of CNQX eliminates mixed PD-SB events but not single isolated SBs, suggesting that the coupling of PD and SB in mixed events is glutamatergic (*Knogler et al., 2014*). Therefore, we aimed to generate a model with the following characteristics: (1) double coils lasting about 1 s in duration accounting for over half of the coiling events, (2) a dependence of double coiling upon excitatory synaptic transmission, (3) an increase in multiple coiling events in the absence of inhibitory synaptic transmission, and (4) the presence of mixed PD-SB events with similar sensitivity to the blockade of excitatory synaptic transmission as double coils.

### Network description

(*Figure 3A*) To implement a model capable of generating double coils that depend upon glutamatergic transmission, we built upon the single coiling model by adding two populations of neurons. We reasoned that if double coiling depended upon excitatory neurotransmission, then a population of commissural excitatory neurons could be necessary to trigger a second contralateral coil in double coils. V0v neurons are a population of glutamatergic commissural INs, some of which may be involved in larger amplitude locomotor movements such as coiling (*Jay and McLean, 2019*). Thus, we added a chain of V0vs (10 neurons for each side) electrically coupled to the previous scaffold (i. e., the ipsilateral IC-MN-V0d scaffold). To generate the crossing excitation underlying the second coil, all V0vs projected glutamatergic synapses to contralateral ICs. Electrical synapses were formed with neighboring MNs, V0ds, and V0vs (the nearest three of each type of neuron in both the rostral and the caudal directions). Ipsilateral ICs were coupled with V0vs in the first four rostral somites.

The second population of neurons that we added was ipsilaterally projecting excitatory neurons present at this stage and shown to receive mixed PD-SB events (*Knogler et al., 2014*). These neurons have been suggested to be circumferential ipsilateral descending neurons that arise from the V2a population. In the model, V2as were electrically coupled to IC neurons and projected glutamatergic synapses to V0vs. This chemical synapse caused a delay after the initiation of the initial coil that facilitates the second contralateral coil (see below). V2as most likely also excite MNs, based on the data from *Knogler et al., 2014*. For computational simplicity, we omitted this connection as it was unnecessary for double coilings to be generated, though this may reduce the amplitude of the

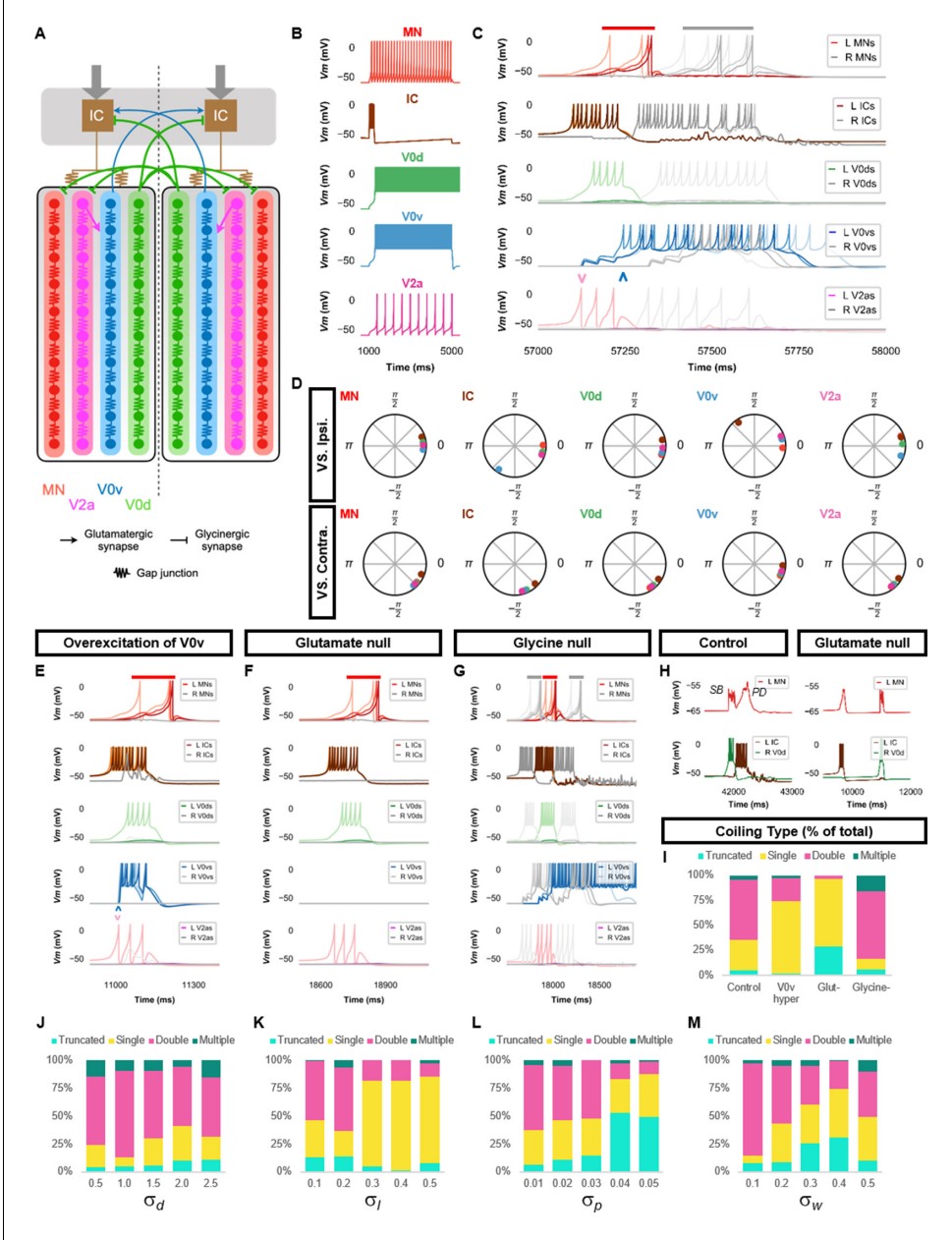

**Figure 3.** Double coiling model relies on a hybrid network of electrical and chemical synapses. (A) Schematic of the double coiling model. Gap junctions between spinal neurons are not depicted. Dashed line indicates the body midline. Gray arrows indicate descending motor command. (B) Membrane potential (*Vm*) response of isolated spinal neuron models to a depolarizing current step. (C) *Vm* of spinal neurons during a double coil. (D) The phase delay of left neurons in relation to ipsilateral and contralateral spinal neurons in the fifth somite and an IC in the rostral kernel during five consecutive left-right double coils. The reference neuron for each polar plot is labeled, and all neurons follow the same color-coding as the rest of the figure. A negative phase delay indicates that the reference neuron precedes the neuron to which it is compared. A phase of 0 indicates that a pair of neurons is in-phase; a phase of π indicates that a pair of neurons is out-of-phase. *Vm* in simulations where (E) the weights of the V2a to V0v and the V0v to IC synapses were increased to show that early excitation of V0v prevented the initiation of a second coil following a single coil, (F) all glutamatergic transmission was blocked, and (G) glycinergic transmission was blocked. (H) *Top row*, mixed event composed of a synaptic burst (SB) directly followed by a periodic depolarization (PD) in a motoneuron in control but not in glutamate null conditions. *Bottom row*, *Vm* in left IC and right V0d during events in top row. (I) Proportions of single, double, multiple, and truncated coiling events under control, glutamate null (Glut⁻), overexcited V0vs (V0v hyper), and glycine null (Glycine⁻) conditions. *Figure 3 continued on next page*

*Figure 3 continued*

Each condition was tested with five 100,000 ms runs with $\sigma_d$ = 0.5, $\sigma_p$=0.01, and $\sigma_w$ = 0.05. (J–M) Sensitivity testing showing proportions of single, double, multiple, and truncated coiling events during ten 100,000 ms runs for each value of $\sigma_d$, $\sigma_l$, $\sigma_p$, and $\sigma_w$ tested. Solid red and gray bars in (C,E–G) indicate the duration of coils. Chevrons in (C and E) denote the initial spiking of V0vs and V2as to indicate latency of V0v firing during the first coil. For (C, E–G), the *Vm* of a rostral (lightest), middle, and caudal (darkest) neuron is shown, except for IC neurons that are all in a rostral kernel. L: left, R: right. See also *Figure 3—figure supplements 1* and *2* and *Figure 3—videos 1–4*. IC, Ipsilateral Cauda; MN, motoneuron.

The online version of this article includes the following video and figure supplement(s) for figure 3:

**Figure supplement 1.** Double coiling model with no V2a to V0v synapses, no contralateral synapses, or with 30 somites.

**Figure supplement 2.** Sensitivity testing of the double coiling model for the glycinergic reversal potential ($E_{gly}$), weights of chemical synapses ($\sigma_{w,\ chem}$), and weights of gap junctions ($\sigma_{w,\ gap}$).

**Figure 3—video 1.** Double coiling model.
https://elifesciences.org/articles/67453#fig3video1

**Figure 3—video 2.** Glutamate null double coiling model.
https://elifesciences.org/articles/67453#fig3video2

**Figure 3—video 3.** Overexcited V0v double coiling model.
https://elifesciences.org/articles/67453#fig3video3

**Figure 3—video 4.** Glycine null double coiling model.
https://elifesciences.org/articles/67453#fig3video4

**Figure 3—video 5.** 30-somite double coiling model.
https://elifesciences.org/articles/67453#fig3video5

coils. As V2as display SBs at this stage, we modeled glycinergic projections from V0ds to contralateral V2as such that the *i*th V0d projected to all contralateral V2as between the $i-5$ and $i+5$ segments like how V0ds project to contralateral MNs.

Left and right ICs received a tonic motor command though we delayed the activation of the tonic command to the right side by 1500 ms to ensure that double coils were not near-coincident bilateral single coils. We modified several parameters of the ICs, most notably increases in the *a* and the *k* parameters, to produce a more extended inter-coiling period (*Figure 3B*) than seen in single coiling (*Knogler et al., 2014*). A reminder that the *a* parameter represents the time scale of the recovery variable *u* that returns the membrane potential to rest. The *k* parameter shapes subthreshold dynamics.

## Simulations results

Simulations of the double coiling model frequently generated pairs of successive, left-right alternating coils lasting about 1 s in total (*Figure 3C*, *Figure 3—video 1*). In five 100,000-ms long runs of the base model with a minimal amount of variability added to several model parameters ($\sigma_d$ = 0.5, $\sigma_p$ = 0.01, and $\sigma_w$ = 0.05), approximately 60% of events were double coils, with the rest mainly being single coils (31%), and very few triple coils or truncated single coils (5% each) (*Figure 3I*).

The timing of ICs, V2as, and V0vs (*Figure 3C*) suggests that double coils were generated by ipsilateral recruitment of V2as and V0vs during the first coil, which led to activation of the contralateral ICs to initiate the second coil. This sequence is supported by an analysis of the phase delays (*Figure 3D*). IC firing precedes the firing of all other ipsilateral spinal neurons suggesting they drive the activity of each coil. V2a activity precedes that of V0vs, which suggests that V2as recruit V0vs. This recruitment of V0vs by V2as is supported by simulations where the V2a to V0v synapse is removed (*Figure 3—figure supplement 1A*). V0v activity succeeds all other ipsilateral spinal INs, suggesting they are the last INs active during the first coil in a double coiling event. A key to generating double coils in our model was thus to delay the activation of V0vs. This delay enabled the activation of contralateral ICs after the first coiling is completed and when commissural inhibition of the contralateral IC has also terminated. If the activation of ipsilateral V0vs occurred too early during the first coiling, which can be produced by increasing the weight of the V2a to V0v and the V0v to contralateral IC glutamatergic synapses, the occurrence of a second coil is less likely (*Figure 3E,I*, *Figure 3—video 2*).

To further underscore the importance of glutamatergic transmission to double coiling as reported experimentally (*Knogler et al., 2014*), blocking glutamatergic transmission in the model greatly reduced the number of double coils (*Figure 3F,I*, *Figure 3—video 3*). On the other hand, blocking glycinergic synapses increased multiple coilings of three or more coils (*Figure 3G,I*, *Figure 3—video 4*) as *Knogler et al., 2014* reported. This effect presumably occurs due to the unopposed reverberating commissural excitation of ICs by V0vs. Indeed, silencing V0v synapses in a model with no glycinergic synapses blocks double and multiple coils (*Figure 3—figure supplement 1B*).

The sequencing of commissural excitation and inhibition in the generation of double coils is further underscored by the presence of mixed SB-PD or PD-SB events (*Figure 3H*) observed experimentally in hyperpolarized MNs (*Knogler et al., 2014*). In these mixed events, the PDs were generated by gap junction coupled ICs during the ipsilateral coil, whereas contralateral V0ds activated during the contralateral coil generated the SBs in the ipsilateral MNs. Blocking glutamatergic transmission in our model uncoupled PDs and SBs (*Figure 3H*) as observed experimentally (*Knogler et al., 2014*).

Just as the robustness of the single coiling model was tested through modifications to the base model and several sensitivity tests, we also tested the robustness of the double coiling model. First, we increased the size of the model from 10 to 30 somites. Modifications of the tonic motor command amplitude, length of IC axons, gap junction coupling from IC to MN, and the synapses from MN to muscle cell, V0v to IC, V2a to V0v enabled the generation of double coils in this model (*Figure 3—figure supplement 1C*, *Figure 3—video 5*). In the 10-somite base model, we also tested the role of the glycinergic reversal potential. The value of this parameter was hyperpolarized from $-45$ mV in the single coiling model to $-58$ mV in the base double coiling model. This shift was intended to reflect gradual hyperpolarization of the reversal potential of glycine during development (*Ben-Ari, 2002*; *Saint-Amant and Drapeau, 2000*, *Saint-Amant and Drapeau, 2001*). We tested the double coiling model at values ranging between $-46$ and $-70$ mV (*Figure 3—figure supplement 2A*). We found that the proportion of double coils seemed to be higher, and the proportion of multiple coils was increased at more depolarized values of the glycinergic reversal potential. The proportion of double coils was relatively constant at more hyperpolarized values of the glycinergic reversal potential.

To test whether the double coiling model was sensitive to within-model parameter variability, we ran sets of ten 100-s long simulations at various values of $\sigma_d$, $\sigma_l$, $\sigma_p$, and $\sigma_w$ (*Figure 3J-M*). Again, we found that relatively small levels of variability in the parameters governing membrane dynamics ($\sigma_p$) decreased the proportion of coiling events that were double coils and increased the number of truncated coils (*Figure 3L*). Moderate levels of variability in the parameters governing axonal length ($\sigma_l$) or synaptic weight ($\sigma_w$) decreased the proportion of double coils while increasing single coils and sometimes truncated coils (*Figure 3K, M*). The proportion of coiling events was largely unaffected by variability in the amplitude of the motor command ($\sigma_d$, *Figure 3J*).

Considering that the generation of double coils was sensitive to chemical synaptic activity and gap junctions (*Knogler et al., 2014*), we tested the sensitivity of the model to variability in the weights of only chemical synapses ($\sigma_{w,chem}$) and only gap junctions ($\sigma_{w,gap}$). We found that the proportion of double coils was relatively more sensitive to the variability of gap junctions than chemical synapses (*Figure 3—figure supplement 2B, C*).

## Generation of swimming pattern by spinal network oscillators (>2–3 dpf)

Around 2 or 3 dpf, zebrafish transition from coiling movements to swimming (*Drapeau et al., 2002*; *Saint-Amant and Drapeau, 1998*). This transition entails two fundamental changes in locomotor movements: first, long, slow coils are replaced by quick, short tail beats; and second, the number of consecutive tail beats are increased from the two side-to-side coilings seen in double coils to multiple consecutive tail beats that compose each swimming episode. One of the emerging swimming movements is beat-and-glide swimming, characterized by short swimming episodes lasting several hundreds of milliseconds separated by gliding pauses and lasting several hundreds of milliseconds (*Budick and O'Malley, 2000*; *Buss and Drapeau, 2001*). Swim episodes consist of repetitive left-right alternating, low-amplitude tail beats that propagate from the rostral toward the caudal end of

the fish body and are generated approximately at 20–65 Hz (*Budick and O'Malley, 2000*; *Buss and Drapeau, 2001*).

Beat-and-glide swimming can be produced in isolated larval zebrafish spinal cord preparations by NMDA application (*Lambert et al., 2012*; *McDearmid and Drapeau, 2006*; *Wiggin et al., 2012*) or by optogenetic stimulation of excitatory spinal neurons (*Wahlstrom-Helgren et al., 2019*). This capacity suggests that the transition from coiling to swimming involves a delegation of rhythm generation from ICs to spinal locomotor circuits (*Roussel et al., 2020*). Therefore, we sought to model a spinal network that could generate beat-and-glide swimming activity hallmarks—swim episodes lasting about 200–300 ms with repeated left-right alternating low-amplitude tail beats at around 20–65 Hz—without relying on pacemaker cells.

Recent experimental studies have also started to delineate the contributions of specific populations of spinal neurons to swimming. Ablation of ipsilaterally projecting, excitatory neurons in the V2a population eliminates swimming activity (*Eklöf-Ljunggren et al., 2012*). Genetic ablation of ipsilaterally projecting, inhibitory neurons in the V1 population affects swim vigor but has no effects on the patterning of swimming (*Kimura and Higashijima, 2019*). Genetic ablation of a subset of commissural inhibitory neurons in the dI6 population reduces left-right alternation (*Satou et al., 2020*). We sought to replicate the role of these neurons in our model.

## Network description

(*Figure 4A*) Whereas coiling is likely to be generated by primary MNs, swimming is more likely to involve secondary MNs (*Ampatzis et al., 2013*; *Liu and Westerfield, 1988*). There are more secondary than primary MNs, and new spinal neurons are born at the same time as secondary MNs (*Bernhardt et al., 1990*). To emulate the increase in the number of spinal neurons that may underlie swimming, we increased the size of the fish from 10 to 15 segments and accordingly increased the number of MNs, V0vs, and V2as from 10 to 15. Thus, each model somite in our swimming model represented two biological somites instead of three in our coiling models. IC neurons were removed from the model to reduce computational load. We are not aware of any experimental evidence of the involvement of IC neurons in later swimming stages.

We introduced two populations of neurons for the beat-and-glide model. Commissural inhibitory CoBL neurons (including neurons from the dI6 and V0d populations) are active during swimming (*Liao and Fetcho, 2008*; *Satou et al., 2020*). V0d neurons were involved with faster swimming, whereas dI6s were more likely to be active during slower swimming (*Satou et al., 2020*). Therefore, we modeled CoBL neurons as dI6s. dI6s thus replaced the V0ds in the coiling models as the source of contralateral inhibition in the swimming model. CoBL neurons have been shown to project to MNs, dI6s, and unidentified ipsilateral descending spinal neurons that could be V2as (*Satou et al., 2020*). We added fifteen dI6s per side. We modeled the projection pattern of dI6s based on the bifurcating axons with short ascending and long descending branches of a subset of neurons in the dI6 subpopulation (*Satou et al., 2020*). Thus, the $i$th dI6s projected ascending branches to their rostral targets in the $i-1$th segment and projected descending branches to their caudal targets between the $i+1$ and $i+3$ segments.

A second new population of neurons was the V1 INs that include circumferential ascending INs that emerge during the second wave of neurogenesis in the spinal cord (*Bernhardt et al., 1990*). While V1 neurons were not included in the coiling models because their role in that form of movement is unclear, experiments in which the genetically identified V1 neurons are ablated suggest a role in controlling swim vigor (*Kimura and Higashijima, 2019*). We thus modeled V1s as a population of tonic firing neurons with ipsilateral ascending glycinergic projections. We added 15 V1s per side. We distributed V1s from segment 2 to the caudal end because our preliminary simulations suggested that starting the distribution of V1s at segment 1 made the episode duration more variable. In our model, V1s project segmental and ascending ipsilateral glycinergic synapses with rostral V2as (*Kimura and Higashijima, 2019*), dI6s, and V0vs (*Sengupta et al., 2021*) such that the V1s in the $i$th segment project to their rostral targets in the $i-1$ to $i-2$ segments. The V1 projections were short based upon recent evidence that their projections to motor circuits are constrained to segmental and immediately neighboring somites (*Sengupta et al., 2021*). Reciprocally, V2as formed glutamatergic synapses to caudally located V1. The reversal potential of the glycinergic synapses from dI6 and V1 neurons was set at −70 mV. This value is hyperpolarized compared to values in the coiling

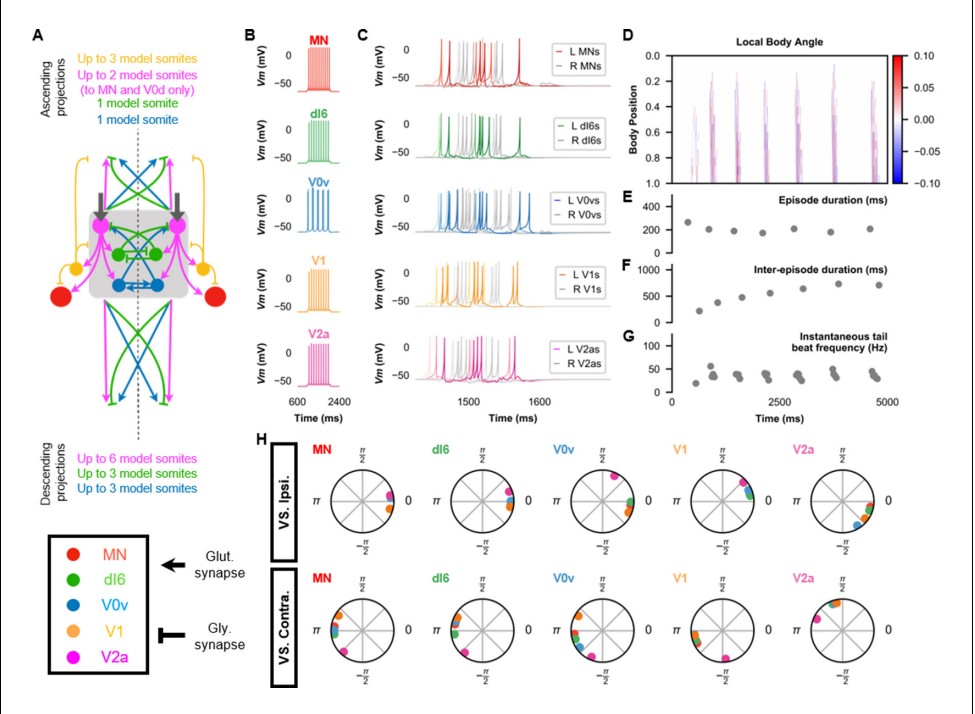

**Figure 4.** The base model for beat-and-glide swimming. (**A**) Schematic of the model architecture underlying beat-and-glide swimming. (**B**) Membrane potential (*Vm*) response to a depolarizing current step of isolated spinal neurons in the model. (**C**) *Vm* of spinal neurons during a beat-and-glide swimming simulation. The *Vm* of a rostral (lightest), middle, and caudal (darkest) neuron is shown. L: left, R: right. (**D**) Heat-map of local body angle. (**E**) Episode duration, (**F**) inter-episode interval, (**G**) instantaneous tail beat frequency, and (**H**) the phase delay of left neurons in relation to ipsilateral and contralateral spinal neurons in the 10th somite during a 10,000 ms simulation. The reference neuron for each polar plot is labeled, and all neurons follow the same color-coding as the rest of the figure. A negative phase delay indicates that the reference neuron precedes the neuron to which it is compared. A phase of 0 indicates that a pair of neurons is in-phase; a phase of π indicates that a pair of neurons is out-of-phase. See also *Figure 4—video 1*. IC, Ipsilateral Caudal; MN, motoneuron.

The online version of this article includes the following video for figure 4:

**Figure 4—video 1.** Beat-and-glide model.

https://elifesciences.org/articles/67453#fig4video1

model considering the known hyperpolarization of the chloride equilibrium potential during development (*Ben-Ari, 2002*), which has also been observed in the zebrafish nervous system (*Zhang et al., 2010*).

V2as were considered the primary source of rhythmogenesis in our models of beat-and-glide swimming based on previous studies showing the necessity and sufficiency of V2a neurons to swimming activity (*Eklöf-Ljunggren et al., 2012*; *Ljunggren et al., 2014*). In the swimming model, V2as projected segmental and descending projections to dI6s, MNs, V0vs, V1s, and caudal V2as (the *i*th V2a projected to all caudal V0vs, dI6s, V2as, and MNs between the *i*+1 and *i*+6 segments). Connections between V2as and V2as (*Ampatzis et al., 2014*; *Menelaou and McLean, 2019*; *Song et al., 2020*) and from V2a neurons to MNs (*Ampatzis et al., 2014*; *Menelaou and McLean, 2019*; *Song et al., 2020*) have been reported. Connections from V2as to dI6s have not been studied yet, and so we based them on the reported connections from V2a neurons to V0d neurons, another population of commissural inhibitory INs (*Menelaou and McLean, 2019*). A subtype of V2a neurons that project to MNs was shown to bifurcate and have short ascending branches (*Menelaou and McLean, 2019*), which we modeled in addition to an ascending V2a to V0v connection (the *i*th V2a projected to rostral V0vs and MNs in the *i*−1 and *i*−2 segments) that remains to be confirmed.

Less is known about the connection patterns of commissural excitatory neurons at larval stages. However, in adult zebrafish, V0v commissural excitatory neurons have been shown to have

bifurcating axons with shorter ascending branches and longer descending branches (*Björnfors and El Manira, 2016*). We modeled V0vs to project only to contralateral V2as. The *i*th V0vs projected ascending branches to their rostral targets in the $i-1$th segment and projected descending branches to their caudal targets between the $i+1$ and $i+3$ segments.

Whether rhythmic motor commands from supraspinal commands generate rhythmic tail beats at the spinal cord level is unclear. There is evidence for rhythmic and tonic activity in reticulospinal neurons involved in swimming (*Kimura et al., 2013*). However, the isolated zebrafish spinal cord can generate rhythmic activity similar to swimming (*McDearmid and Drapeau, 2006*; *Wahlstrom-Helgren et al., 2019*; *Wiggin et al., 2012*; *Wiggin et al., 2014*). Therefore, in our model, V2as received a tonic motor command in the form of a DC current to test whether the rhythm and pattern of swimming could be generated solely from the activity of spinal circuits.

Most spinal neurons at larval stages exhibit either tonic firing or firing with spike rate adaptation (*Kimura and Higashijima, 2019*; *Menelaou and McLean, 2012*; *Menelaou and McLean, 2019*; *Satou et al., 2020*), while a subset of MNs showing intrinsic burst firing (*Menelaou and McLean, 2012*). Therefore, we posited that the generation of rhythmic tail beats was not dependent upon the presence of intrinsically bursting neurons. Almost all of the neurons in the beat-and-glide swimming model were modeled to fire tonically (*Figure 4B*). Our base model was able to generate the beat-and-glide swimming pattern—alternating episodes of tail beats followed by silent inter-episode intervals each lasting hundreds of seconds—if V0vs were modeled to exhibit a more chattering or bursting firing pattern (*Figure 4B*).

As the model is symmetrical, including the motor command, we found that the model produces synchronous left-right activity unless we introduced some variability in commissural connections. With no a priori knowledge of where such variability could arise from, we chose to introduce a small amount of variability in the contralateral inhibition of dI6s by dI6s. The synaptic weights of this connection were scaled by a random number picked from a Gaussian distribution with mean =1, and standard deviation of 0.1, and this was sufficient to generate alternating left-right alternation. We did not seek to further characterize the variability required to generate left-right alternation.

## Simulation results

The beat-and-glide swimming model exhibited short-duration (hundreds of milliseconds) swimming episodes with left-right alternation and tail beat frequencies between 20 and 60 Hz (*Figure 4C–G*, *Figure 4—video 1*). The characteristics of the swimming episodes in our simulations were close to those described for free swimming in larval zebrafish by *Buss and Drapeau, 2001*, though the swimming output in our simulations had larger episode durations, shorter inter-episode intervals, and lower tail beat frequencies (*Table 5*).

An analysis of the phase delay between neuron populations during beat-and-glide swimming in the base model shows that the activity of ipsilateral, glutamatergic V2as precedes the activity of all ipsilateral neurons (*Figure 4H*). This earlier firing of V2as suggests that these spinal neurons drive the activity of the ipsilateral spinal swimming circuit. On the other hand, the glycinergic V1 neurons succeed all ipsilateral spinal neurons, suggesting they provide negative feedback to ipsilateral spinal swimming circuits. A hyperpolarized reversal potential of glycinergic synapses at this developmental stage strengthens the negative feedback (see Sensitivity testing to different values of the glycinergic reversal potential below). The contralateral dI6s and V0vs are out-of-phase with contralateral spinal neurons, consistent with their role in mediating left-right coordination. The longest delay between

**Table 5.** Comparison of beat-and-glide swimming in model and experimental data from *Buss and Drapeau, 2001*.

| Parameter | Base model | *Buss and Drapeau, 2001* |
|---|---|---|
| Mean episode duration (ms) | 234±6 | 180±20 |
| Mean inter-episode interval (ms) | 242±20 | 390±30 |
| Mean tail beat frequency (Hz) | 30.0±0.6 | 35±2 |

Values in mean± standard error. n=ten 10,000 ms-long simulations for the base model, and n=12 animals for the data from *Buss and Drapeau, 2001*.

V2as and their ipsilateral counterparts was with the V0vs, which is reminiscent of V2a firing preceding V0v firing in the double coiling model to ensure a sufficient delay for the initiation of the second coil. Thus, the generation of alternating left-right tail beats would seem to require a certain delay in the excitation of contralateral swimming circuits.

To further investigate the role of specific neurons in the model's swimming activity, we performed simulations composed of three 5000 ms long epochs: epoch 1, where the model was intact; epoch 2, where we silenced the targeted neurons by removing their synaptic inputs; and epoch 3, where the synaptic inputs to the targeted neurons were restored. Silencing V2as abolished the generation of tail beats (*Figure 5A–F*, *Figure 5—figure supplement 1A–F*), underscoring their primacy to the generation of tail beats (*Eklöf-Ljunggren et al., 2012*; *Ljunggren et al., 2014*). Commissural excitation mediated by V0vs seemed to be very important in maintaining the beat-and-glide pattern. Silencing V0vs diminished but did not eliminate the rhythmic firing of V2as or MNs. During epoch 2, the tonic motor command continued to activate V2as. Pairs of left-right tail beats may result from the commissural inhibition by dI6s that was still present. However, removing the contralateral excitation by V0v prevented the repetitive activation of the silent side after each tail beat, which severely reduced episode duration and the number of tail beats generated in each episode (*Figure 5G–L*, *Figure 5—figure supplement 1G–L*, *Figure 5—video 1*). These simulations suggest that commissural excitation is necessary to repeatedly activate the silent contralateral side during ongoing swimming to ensure successive left-right alternating tail beats and longer swim episodes (*Björnfors and El Manira, 2016*; *Saint-Amant, 2010*).

Genetic ablation of the ipsilaterally ascending inhibitory V1 INs increased swim vigor but produced no overt changes in swimming patterns (*Kimura and Higashijima, 2019*). Consistent with those experimental results, simulating the removal of ipsilateral ascending inhibition by silencing V1 INs seemed to increase the amplitude of MN activity (*Figure 6A,B*, *Figure 6—figure supplement 1A–F*, *Figure 6—video 1*). While the overall beat-and-glide pattern persisted, the duration of episodes was increased, and inter-episode intervals were shortened between epochs 1 and 2 (*Figure 6C,D*). Tail beat frequency was increased (*Figure 6E*). Left-right alternation was reduced in these simulations and during simulations where dI6s were silenced (*Figure 6F,G,L*, *Figure 6—figure supplement 1G–L*). The reduction in left-right alternation during simulations where dI6s were silenced was greater in caudal somites than in rostral somites (*Figure 6L*). Note that while left-right alternation was reduced, this did not preclude left-right alternating tail beats from being generated (*Figure 6—video 2*). The reduction of left-right coordination seen here was comparable to levels seen after genetic ablation of a commissural inhibitory subpopulation of dI6 INs (*Satou et al., 2020*) but is not sufficient to prevent left-right alternation. Since swimming is generated by rostrocaudal propagation of contractile waves, any left-right alternation in rostral segments will inevitably sway the rest of the body, as suggested by the musculoskeletal model. The precise kinematics of the tail beats will be affected by the reduction of left-right alternation observed (*Figure 6—figure supplement 2*, *Figure 6—video 2*). Finally, silencing dI6s had negligible effects on the episode duration and inter-episode interval but increased tail beat frequency when comparing epochs 1 and 2 and may increase the amplitude of motor activity (*Figure 6H–K*).

Previous experimental results showed that strychnine application disturbed swimming in the 20–40 Hz range at later stages of development (*Roussel et al., 2020*). Our model's behavior to loss of glycinergic transmission was tested. Removal of all glycinergic transmission led to continual tail beats with minimal gliding periods (*Figure 7*). Motor output was increased during the epoch of no glycinergic transmission (*Figure 7B*), episode duration was considerably lengthened, and the inter-episode interval was shortened (*Figure 7C–D*). The frequency of tail beats increased (*Figure 7E*). Left-right alternation was reduced, particularly at caudal somites (*Figure 7F*), which did not preclude left-right tail beats but altered swimming kinematics (*Figure 7—figure supplement 1*, *Figure 7—video 1*). These results indicate that removing glycinergic transmission in the model led to near-continuous swimming activity with altered kinematics and greater frequencies of tail beats.

We then proceeded to test the sensitivity of the base model to several model parameters. Since some V2a INs in adult zebrafish have pacemaker capacities (*Song et al., 2020*), we tested whether we could also generate beat-and-glide swimming in a model with bursting V2a (*Figure 8A*). A model with bursting V2as where we also decreased the strength of synapses from V2as to other neurons and increased the connection strength of V0vs and dI6s to contralateral V2as (*Table 3*) generated 100–400 ms swimming episodes of left-right alternating tail beats at frequencies around 20–80 Hz

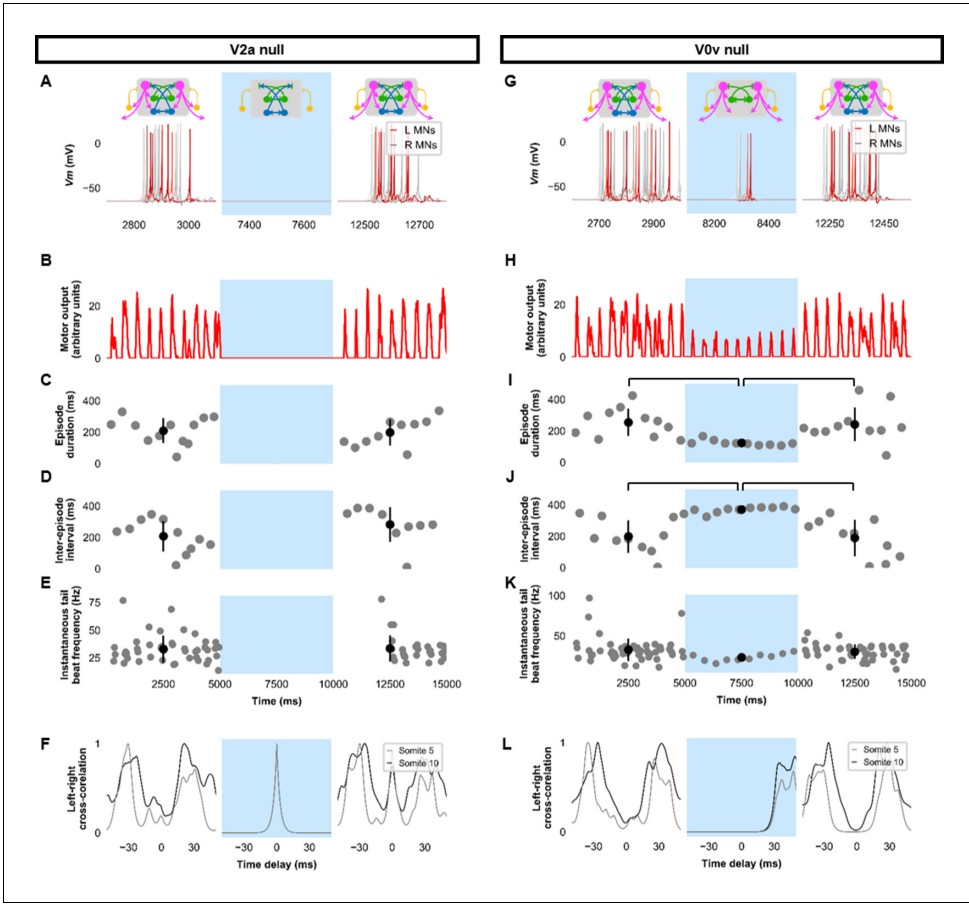

**Figure 5.** Silencing spinal excitatory neurons during beat-and-glide swimming. Simulations consisted of three 5,000 ms epochs. In the middle epoch, silencing of targeted spinal neurons was achieved by removing all synaptic and external currents from the targeted population. Synaptic and external currents were restored in the last epoch. (A–F) Simulations where V2as were silenced and (G–L), where V0vs were silenced. (A, G) *Top*, the functional state of the spinal network during the three epochs. *Bottom*, motoneuron (MN) membrane potential (*Vm*) during simulations where targeted neurons were silenced in the middle epoch. The *Vm* of a rostral (lightest), middle, and caudal (darkest) neuron is shown. (B, H) The integrated muscle output, (C, I) episode duration, (D, J) inter-episode intervals, and (E, K) instantaneous tail beat frequency during each respective simulation. Averages within epoch are shown in black (mean±s.d.). Brackets denote significant pairwise differences. (F, L) The left-right coordination of somites 5 and 10. L: left, R: right. The first part of epoch 3 of the V2a silenced simulation involved synchronous left-right activity, hence the lack of instantaneous tail beat frequency values. *Statistics*: For (C–E), there were no episodes during epoch 2. There were no statistically significant differences between epochs 1 and 3 for any of the parameters. (I) $F_{2,31}=7.2$, $p=0.0029$. (J) $F_{2,28}=10.2$, $p=0.001$. (K) $F_{2,115}=3.0$, $p=0.055$. *P*-values for *t*-tests are found in *Figure 5—source data 1*. See also *Figure 5—figure supplement 1* and *Figure 5—videos 1* and *2*. MN, motoneuron.

The online version of this article includes the following video, source data, and figure supplement(s) for figure 5:

**Source data 1.** P-values for V2a null and V0v null simulations.

**Figure supplement 1.** Membrane potential (*Vm*) of spinal neurons during simulations of beat-and-glide swimming where excitatory neurons were silenced.

**Figure 5—video 1.** V2a knockout beat-and-glide model.
https://elifesciences.org/articles/67453#fig5video1

**Figure 5—video 2.** V0v knockout beat-and-glide model.
https://elifesciences.org/articles/67453#fig5video2

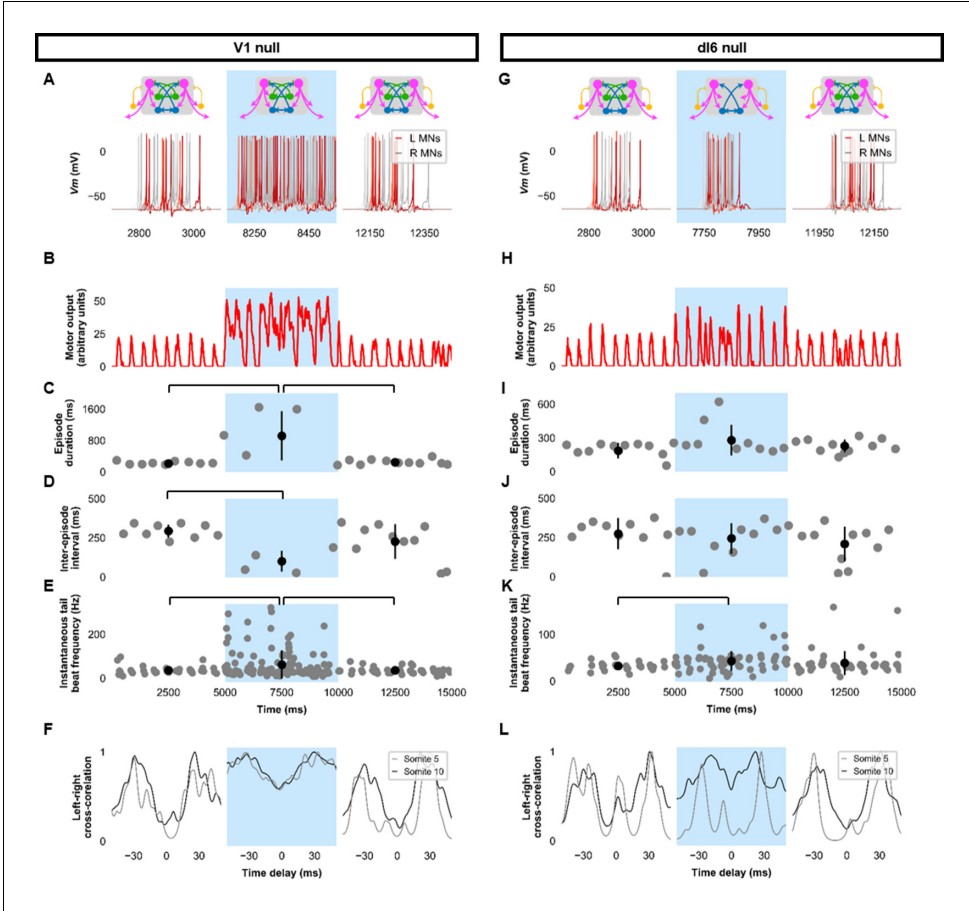

**Figure 6.** Silencing spinal inhibitory neurons during beat-and-glide swimming. Simulations consisted of three 5000 ms epochs. In the middle epoch, silencing of targeted spinal neurons was achieved by removing all synaptic and external currents from the targeted population. Synaptic and external currents were restored in the last epoch. (A–F) Simulations where V1s were silenced and (G–L), where dI6s were silenced. (A, G) *Top*, the functional state of the spinal network during the three epochs. *Bottom*, motoneuron (MN) membrane potential (*Vm*) during simulations where targeted neurons were silenced in the middle epoch. The *Vm* of a rostral (lightest), middle, and caudal (darkest) neuron is shown. (B, H) The integrated muscle output, (C, I) episode duration, (D, J) inter-episode intervals, and (E, K) instantaneous tail beat frequency during each respective simulation. Averages within epoch are shown in black (mean±s.d.). Brackets denote significant pairwise differences. (F, L) The left-right coordination of somites 5 and 10. L: left, R: right. *Statistics*: (C) $F_{2,25}$=10.5, *p*=5.8×10$^{-4}$. (D) $F_{2,22}$=6.6, *p*=0.0063. (E) $F_{2,214}$=6.9, *p*=0.0013. (I) $F_{2,31}$=2.5 *p*=0.10. (J) $F_{2,28}$=0.9, *p*=0.42. (K) $F_{2,145}$=3.5, *p*=0.033. *P*-values for *t*-tests are found in *Figure 6—source data 1*. See also *Figure 6—figure supplements 1* and *2* and *Figure 6—videos 1* and *2*. The online version of this article includes the following video, source data, and figure supplement(s) for figure 6:

**Source data 1.** P-values for V1 null and dI6 null simulations.
**Figure supplement 1.** Membrane potential (*Vm*) of spinal neurons during simulations of beat-and-glide swimming where inhibitory neurons were silenced.
**Figure supplement 2.** Altered kinematics during silencing of dI6 neurons.
**Figure 6—video 1.** V1 knockout beat-and-glide model.
https://elifesciences.org/articles/67453#fig6video1
**Figure 6—video 2.** dI6 knockout beat-and-glide model.
https://elifesciences.org/articles/67453#fig6video2

interspersed by 100–400 ms long silent inter-episode intervals (*Figure 8B–F*, *Figure 8—video 1*). Surprisingly, a model with only tonic firing neurons (*Figure 8G*) was able to generate the hallmarks of beat-and-glide swimming as well (*Figure 8H–L*, *Figure 8—video 2*). While there were eventually longer episodes with shorter inter-episode intervals after 6000 ms, this simulation suggests that the

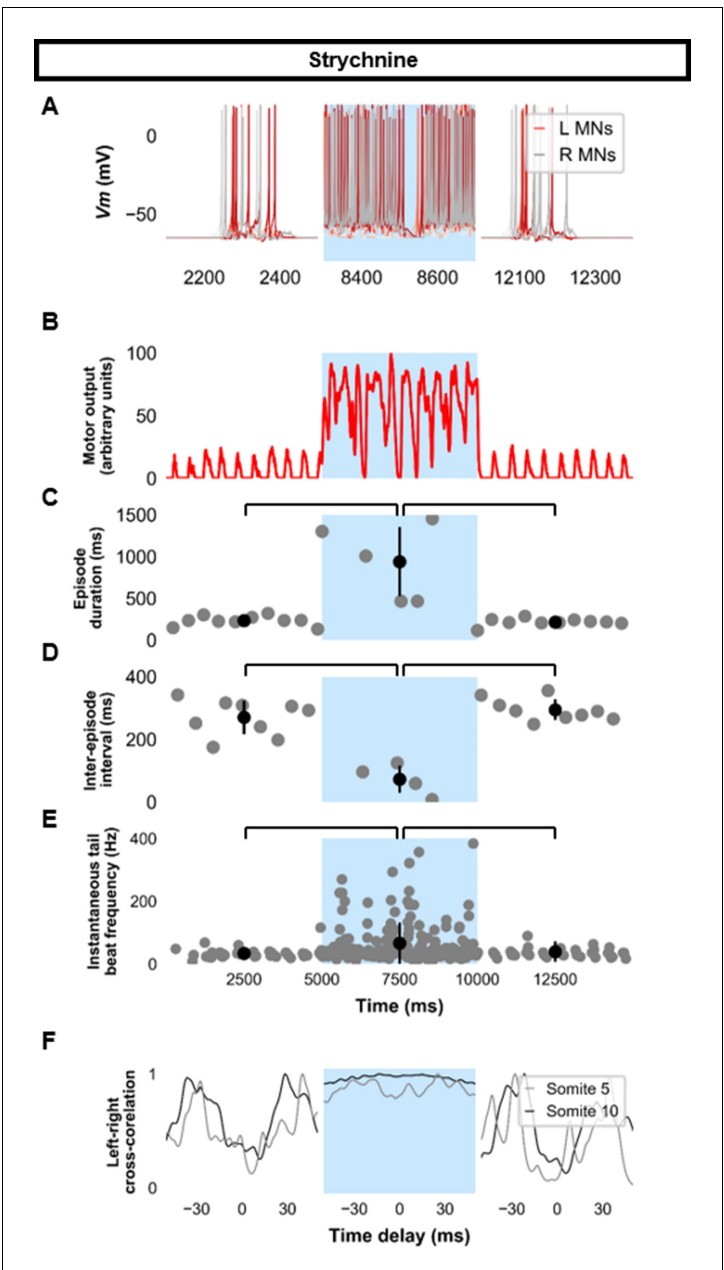

**Figure 7.** Simulating the effects of strychnine on beat-and-glide swimming. Simulations to assess the effects of blocking glycinergic transmission consisted of three 5000 ms epochs. In the middle epoch, all glycinergic currents were blocked. Glycinergic transmission was restored in the last epoch. (A) Motoneuron (MN) membrane potential (*Vm*) during simulations where glycinergic transmission was blocked in the middle epoch. The *Vm* of a rostral (lightest), middle, and caudal (darkest) neuron is shown. (B) The integrated muscle output, (C) episode duration, (D) inter-episode intervals, and (E) instantaneous tail beat frequency during this simulation. Averages within epoch are shown in black (mean±s.d.). (F) The left-right coordination of somites 5 and 10. L: left, R: right. *Statistics*: (C) $F_{2,24}$=2.5, $p$=2.2×$10^{-6}$. (D) $F_{2,21}$=32.0, $p$=8.3×$10^{-7}$. (E) $F_{2,267}$=8.3, $p$=0.0003. *P*-values for *t*-tests are found in *Figure 7—source data 1*. See also *Figure 7—figure supplement 1* and *Figure 7—video 1*.

The online version of this article includes the following video, source data, and figure supplement(s) for figure 7:

**Source data 1.** P-values for strychnine simulations.

**Figure supplement 1.** Altered kinematics during strychnine.

**Figure 7—video 1.** Glycine null beat-and-glide model.

https://elifesciences.org/articles/67453#fig7video1

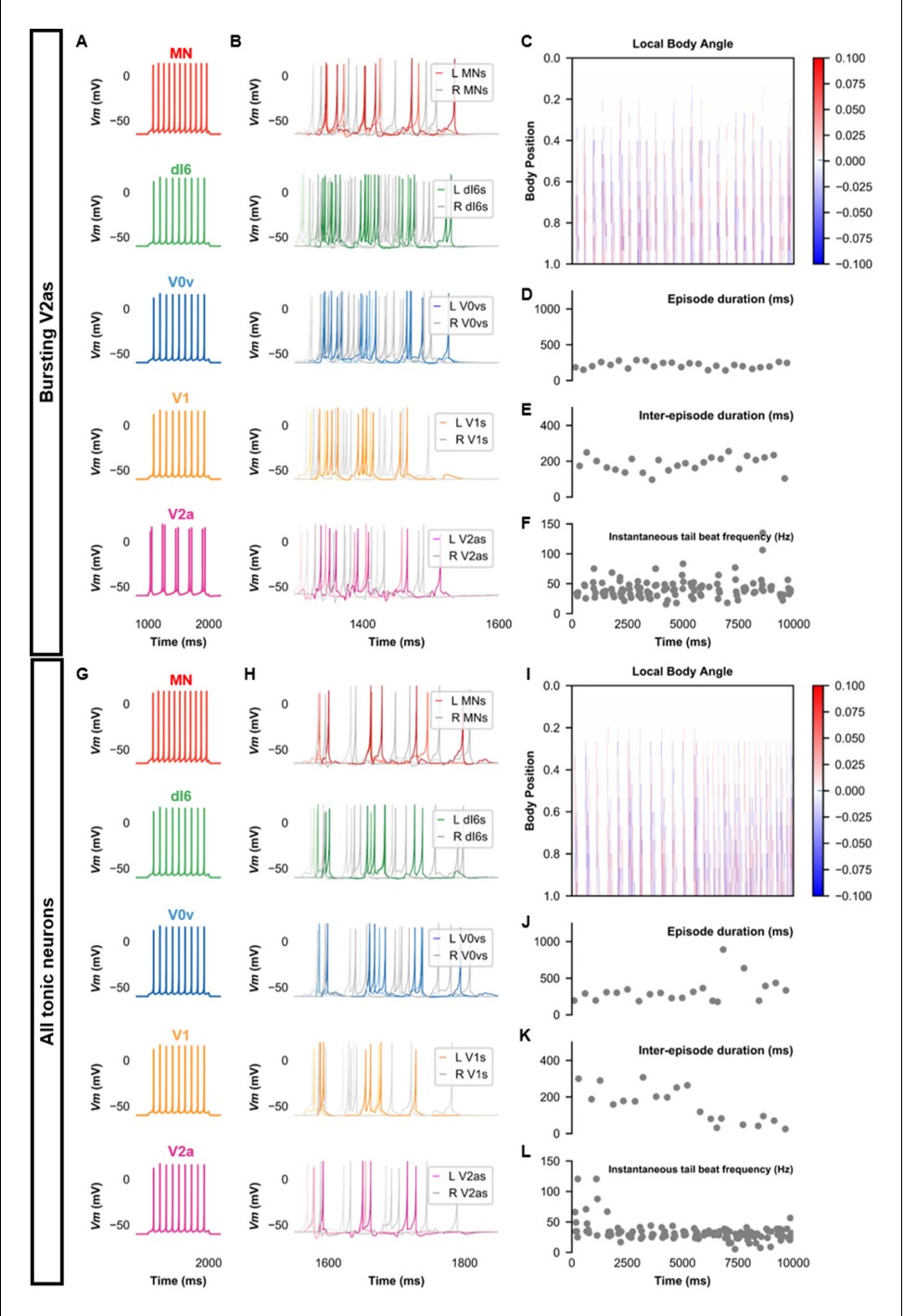

**Figure 8.** Beat-and-glide models with bursting V2a (**A–F**) or only tonic neurons (**G–L**). (**A, G**) Membrane potential (*Vm*) response of isolated neurons in the model to a current step. (**B, H**) *Vm* of spinal neurons during swimming simulation. The membrane potential of a rostral (lightest), middle, and caudal (darkest) neuron is shown. L: left, R: right. (**C, I**) Heat-map of local body angle. (**D, J**) Episode duration, (**E, K**) inter-episode interval, and (**F, L**) instantaneous tail beat frequency during the same simulations as (**B** and **H**), respectively. See also *Figure 8—figure supplements 1* and *2* and *Figure 8—videos 1* and *2*. MN, motoneuron.

The online version of this article includes the following video, source data, and figure supplement(s) for figure 8:

**Source data 1.** P-values for sensitivity testing to values of E_gly.

**Figure supplement 1.** Beat-and-glide swimming model with different number of somites.

**Figure supplement 2.** Sensitivity of beat-and-glide swimming to variability in glycinergic reversal potential ($E_{gly}$).

*Figure 8 continued on next page*

*Figure 8 continued*
**Figure 8—video 1.** Beat-and-glide with bursting V2a model.
https://elifesciences.org/articles/67453#fig8video1
**Figure 8—video 2.** Swimming model with only tonic neurons.
https://elifesciences.org/articles/67453#fig8video2
**Figure 8—video 3.** 30-somite beat-and-glide model.
https://elifesciences.org/articles/67453#fig8video3

architecture of the network is sufficient to generate beat-and-glide swimming for long periods despite the absence of any neurons with bursting properties.

We could generate beat-and-glide swimming in a 30-somite and a 10-somite model (*Figure 8—figure supplement 1*, *Figure 8—video 3*). We also examined the effects of changing the glycinergic reversal potential that was set at −70 mV for the base beat-and-glide model (*Figure 8—figure supplement 2*). We tested the model with $E_{gly}$ values between −56 mV, near the value of $E_{gly}$ in the double coiling model, and −72 mV (*Figure 8—figure supplement 2A,B*). The lower bound of −72 mV is close to the value of $E_{GABA}$ reported in zebrafish retinal ganglion cells around 3–4 dpf (*Zhang et al., 2010*). Episode duration was increased, and inter-episode intervals decreased at more depolarized values of the glycinergic reversal potential leading to the loss of the beat-and-glide pattern (*Figure 8—figure supplement 2C*). Tail beat frequency also increased as glycinergic currents decreased at depolarized glycinergic reversal potentials, and left-right alternation was replaced by left-right synchrony (*Figure 8—figure supplement 2D*).

Finally, the sensitivity of the base model to variability was tested by running sets of ten 10,000-ms long simulations at various values of $\sigma_d$, $\sigma_l$, $\sigma_p$, and $\sigma_w$. We also performed ten 10,000-ms long simulations of the base model (a reminder that there is a random scaling factor to the dI6 to contralateral dI6 synapse in the base model). The episode duration, inter-episode interval, average tail beat frequency in each episode, and the minimum coefficient of the cross-correlation of the left and right muscle output were analyzed (*Figures 9* and *10*). Increases in variability in the motor command drive ($\sigma_d$) seemed to affect inter-episode intervals and left-right alternation but not episode duration and tail beat frequency (*Figure 9A-D*). At similar levels of variability in rostrocaudal axonal length ($\sigma_l$), all four measures of swimming activity were perturbed (*Figure 9E-H*). On the other hand, variability in synaptic weights ($\sigma_w$) affected only episode duration and inter-episode duration (*Figure 9I-L*).

Smaller variability in the parameters shaping the membrane potential dynamics ($\sigma_p$) disrupted the beat-and-glide pattern with no beat-and-glide swimming observed at some values of $\sigma_p$ (*Figure 10A, B*). As $\sigma_p$ was increased, the episode duration, inter-episode interval, and tail beat frequency were disrupted (*Figure 10C, D*). Even slight variability in the parameters shaping membrane potential dynamics (e.g., $\sigma_p$=0.01) resulted in changes in membrane excitability and, in some cases, firing patterns as evidenced by the conversion of some V0vs from burst to tonic firing (*Figure 10E*). Thus, the beat-and-glide model was most susceptible to variations in the parameters determining membrane potential dynamics and similarly sensitive to the other parameters tested.

## Discussion

To our knowledge, this study presents some of the first models of spinal locomotor circuits in developing zebrafish. We have built several spinal locomotor circuit models that generate locomotor movements of the developing zebrafish (*Figure 11*, *Figure 11—video 1*). These models support mechanisms of network operation of developing zebrafish spinal locomotor circuits described experimentally. Our models suggest that the circuitry driving locomotor movements could switch from a pacemaker kernel located rostrally during coiling maneuvers to network-based spinal circuits during swimming. Results from simulations where populations of spinal neurons are silenced were consistent with experimental studies. Our sensitivity analysis suggests that the correct operation of spinal circuits for locomotion is not immune to variations in firing behaviors, length of axonal projections,

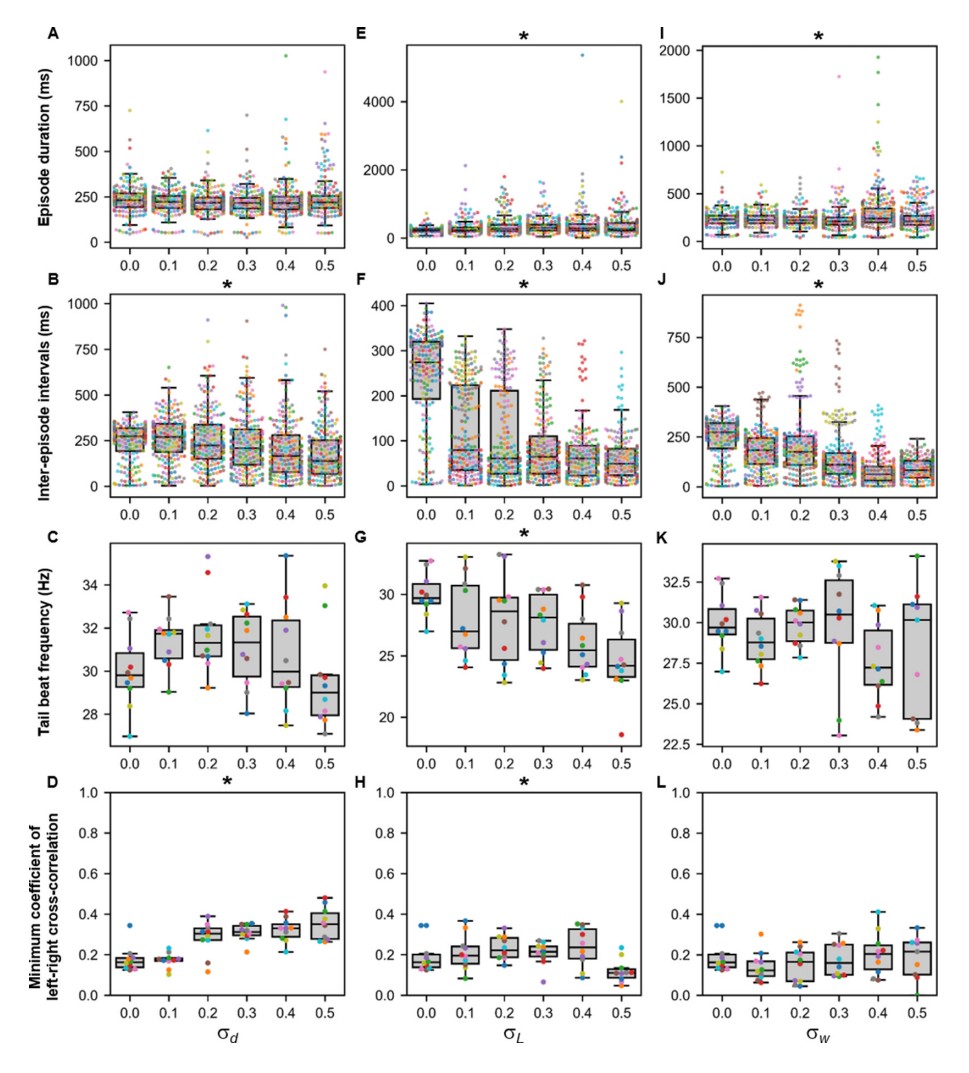

**Figure 9.** Sensitivity of beat-and-glide swimming to tonic motor command amplitude, length of rostrocaudal projections, and synaptic weighting. Ten 10,000-ms long simulations were run for each value of $\sigma_d$ (A–D), $\sigma_L$(E–H), and $\sigma_w$(I–L) tested. (A, E, I) Episode duration. (B, F, J) Inter-episode interval. (C, G, K) Average tail beat frequency during each swimming episode. (D, H, L) Minimum coefficient of the cross-correlation of left and right muscle. The minimum was taken between −10 and 10 ms time delays. Each circle represents a single swimming episode (A, E, I), inter-episode interval (B, F, J), or a single run (all other panels). Each run is color-coded. Runs with only one side showing activity are not depicted in (D and H). Asterisks denote significant differences detected using a one-factor ANOVA test. *Statistics*: (A) $F_{5,1253}$=2.5, $p$=0.03. (Note that there were no pairwise differences detected). (B) $F_{5,1253}$=11.2, $p$=1.3×10$^{-10}$. (C) $F_{5,54}$=1.9, $p$=0.11. (D) $F_{5,54}$= 14.5, $p$=5.2×10$^{-9}$. (E) $F_{5,1253}$=8.7, $p$=3.8×10$^{-8}$. (F) $F_{5,1253}$=118.1, $p$=2.0×10$^{-102}$. (G) $F_{5,54}$=4.0, $p$=0.004. (H) $F_{5,54}$=3.2, $p$=0.014. (I) $F_{5,1400}$=13.5, $p$=6.8×10$^{-13}$. (J) $F_{5,1400}$=74.5, $p$=2.5×10$^{-69}$. (K) $F_{5,53}$=1.3, $p$=0.30. (L) $F_{5,53}$=0.8, $p$=0.55. *P*-values for *t*-tests are found in *Figure 9— source data 1*.

The online version of this article includes the following source data for figure 9:

**Source data 1.** P-values for sensitivity testing to values of sigma D, L and W.

motor command amplitude, and synaptic weighting. However, the sensitivity to these parameters is variable.

## Pacemaker-based network for early behaviors

The earliest locomotor behaviors in zebrafish, namely single and multiple coilings, require global recruitments of neurons to synchronously contract all ipsilateral muscles (*Warp et al., 2012*).

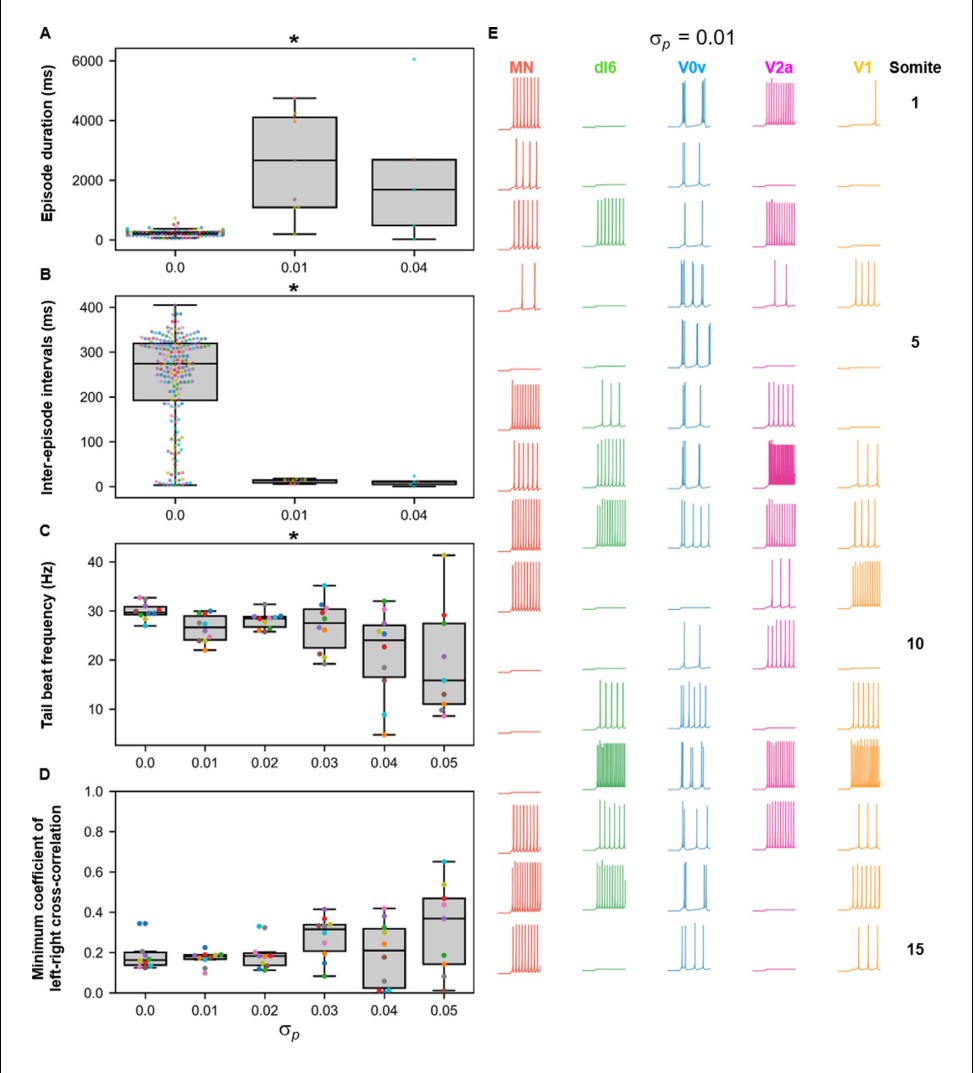

**Figure 10.** Sensitivity of beat-and-glide swimming to variability in membrane potential dynamics. Ten 10,000-ms long simulations were run at each value of $\sigma_p$ (A–D). (A) Episode duration. (B) Inter-episode interval. (C) Average tail beat frequency during each swimming episode. (D) Minimum coefficient of the cross-correlation of left and right muscle. The minimum was taken between −10 and 10 ms time delays. Each circle represents a single swimming episode (A), inter-episode interval (B), or a single run (C, D). Each run is color-coded. Runs not depicted exhibited either continual motor activity with no gliding pauses or no swimming activity. Asterisks denote significant differences detected using a one-factor ANOVA test. (E) Responses to a 1-s long step current of all neurons on the left side in a model where $\sigma_p$=0.01. Step current amplitudes varied between populations of neurons. The amplitude of the step currents to each population is the same as in *Figure 4B*. The simulation of the model with these neurons generated continued swimming activity with no gliding pauses. The neurons are ordered by somite, from somite 1 at the top to somite 15 at the bottom. *Statistics*: (A) $F_{2,211}$=143.8, $p$=4.0×10$^{-40}$. (B) $F_{2,211}$=32.3, $p$=5.8×10$^{-13}$. (C) $F_{5,53}$=4.0, $p$=0.0036. (D) $F_{5,53}$=2.1, $p$=0.085. P-values for $t$-tests are found in *Figure 10—source data 1*. MN, motoneuron.

The online version of this article includes the following source data for figure 10:

**Source data 1.** P-values for sensitivity testing to values of Sigma P.

Electrical coupling, which lacks the delays inherent with chemical neurotransmission, enables these types of ballistic movements. Early locomotor behavior in zebrafish seems to rely on this architecture, as demonstrated by the necessity of electrical but not chemical synapses (*Saint-Amant and Drapeau, 2000*; *Saint-Amant and Drapeau, 2001*). The rapid and multidirectional current transmission supported by electrical synapses is a perfect solution for en masse activation of a neural circuit

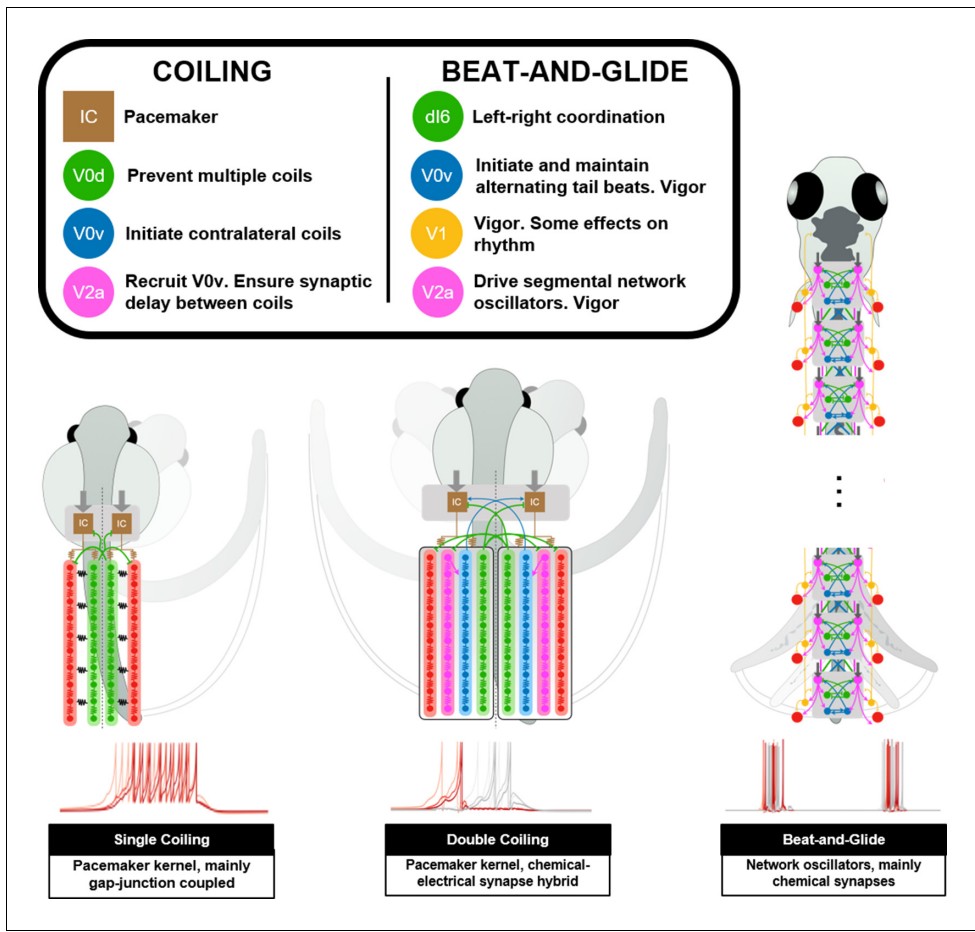

**Figure 11.** Summary figure of computational models of zebrafish locomotor movements during development. See also *Figure 11—video 1*.

The online version of this article includes the following video for figure 11:

**Figure 11—video 1.** Summary of developmental zebrafish computational models.

https://elifesciences.org/articles/67453#fig11video1

(*Bennett and Zukin, 2004*). However, synchronous activation of an ensemble of neurons does not accommodate rhythmic activity, which requires more precise timing and connection strength. For example, the emergence of double coiling in our model was generated by chemical synaptic excitation of contralateral pacemaker neurons that had to be sufficiently delayed to enable the first coil to complete before initiating the second contralateral coil. Commissural glycinergic transmission was also required to tamper down coiling events with more than two successive coils. Suppose multiple coiling is a preparatory stage toward the emergence of repetitive, left-right alternating tail beats. In that case, the possible importance of contralateral excitation and inhibition at this stage presages the establishment of similar operational mechanisms to the generation of swimming.

## Network oscillators for swimming movements

To generate swimming, we delegated the generation of the rhythm driving tail beats to network oscillators distributed along the length of the spinal cord. Spinal locomotor circuits may transition away from pacemakers as the source of the rhythm to prevent being vulnerable to any flaws in the function of a small population of neurons. Also, there may be multiple local rhythms that control

body oscillations along the developing zebrafish's length. Indeed, locomotor output has proven to be very robust to the sectioning of the spinal cord, leading to the suggestion that redundant rhythm-generating circuits must be present within the spinal cord (*McDearmid and Drapeau, 2006*; *Wiggin et al., 2012*; *Wiggin et al., 2014*). Experimental evidence from our lab further suggests that a transition from a rhythm driven by a pacemaker kernel to a rhythm driven by local network oscillators occurs progressively from the caudal toward the rostral end of the body (*Roussel et al., 2020*).

The V2as are well recognized as the neural engine that drives swimming activity in zebrafish spinal circuits (*Eklöf-Ljunggren et al., 2012*; *Ljunggren et al., 2014*). While some V2a INs have shown intrinsic burst firing in the adult zebrafish (*Song et al., 2018*; *Song et al., 2020*), V2a INs in developing zebrafish show either tonic or modestly spike adapting firing (*Menelaou and McLean, 2019*). We thus sought to generate beat-and-glide swimming with tonically firing V2as. Successive left-right alternating tail beats were generated by combining contralateral excitation from bursting commissural excitatory neurons to initiate alternating tail beats and contralateral inhibition to prevent co-contraction of both sides. In fact, a simulation with only tonic firing neurons could also generate beat-and-glide swimming over several seconds. Thus, V2as could very well drive rhythmic tail beats in larval zebrafish while firing tonically. If this is the case, then the central role of V2as depends less on their ability to produce a bursting rhythm. Instead, the pivotal role of V2as in enabling swimming activity would be to coordinate the many spinal IN populations that generate the patterns of repetitive, left-right alternating tail beats seen in developing zebrafish swimming (*Saint-Amant, 2010*). The observation that in our beat-and-glide simulation, V2a neuron firing phasically precedes firing of all the other intrasegmental spinal INs and MNs reinforces the central role of these neurons in driving rhythmic tail beats.

We did find that in simulations where there were only tonic firing neurons, the stability of swimming episode durations started degrading after about 6000 ms. Therefore, burst firing neurons may help to promote the stability of the beat-and-glide pattern. Whether or not this is the case remains to be tested experimentally. Neuromodulation may serve as a mechanism that permits V2as, or other spinal neurons, to toggle between tonic and burst firing through neuromodulation. It is well established that neuromodulators shape the activity of spinal locomotor circuits, likely by regulating intrinsic properties of spinal neurons and through modulation of synaptic weighting and other mechanisms. Blocking D4 dopamine receptors at 3 dpf prevents the transition from burst to beat-and-glide swimming (*Lambert et al., 2012*), suggesting that dopamine from supraspinal sources plays a role in setting the beat-and-glide phenotype by shortening swimming episode duration. Paired recordings of diencephalospinal dopaminergic neurons and spinal MNs during swimming show that these two populations often burst together (*Jay et al., 2015*). Later in development at 6–7 dpf, activation of D1 dopamine receptors increases the recruitment of slow MNs to increase swimming speed (*Jha and Thirumalai, 2020*). The neuromodulator serotonin (5-HT) has been found to either increase motor output by decreasing inter-episode intervals in intact larval zebrafish (*Brustein and Drapeau, 2005*; *Brustein et al., 2003*) or decrease swimming frequency or burst firing in spinalized larvae and adult zebrafish (*Gabriel et al., 2009*; *Montgomery et al., 2018*). In the adult zebrafish, serotonin strengthens inhibition to MNs between tail beats and slows down the onset of the depolarization that initiates each successive tail beat (*Gabriel et al., 2009*). Our model could identify possible targets within the spinal cord for specific neuromodulators of locomotor function in zebrafish.

## Modeling considerations

Our sensitivity analysis suggests that the neuromodulation of intrinsic properties that affect the membrane potential dynamics of spinal neurons could easily modulate locomotor output. The behavior of our models was also sensitive to a lesser degree to increasing variability in descending drive, synaptic weighting, and rostrocaudal extent of connections. Variability in these parameters could change the proportions of coiling types or the values of the characteristics of swimming output measured (e.g. episode duration and inter-episode interval). Model parameter variability sometimes increased the variability of motor output (e.g., *Figure 10*), perhaps indicating a breakdown of the model. However, variability in both model parameters and motor output should not necessarily be considered weaknesses of the model but may instead reflect true biological variability (*Marder and Taylor, 2011*). For instance, recordings of swimming characteristics such as episode duration and inter-episode intervals in larval zebrafish show appreciable variation (*Brustein and Drapeau, 2005*;

*Buss and Drapeau, 2001*). Quantifying heterogeneity within and between animals may guide the appropriate levels of parameter variability to include in future iterations of our models.

Many computational models have already been made of spinal circuits for swimming in species that use undulatory movements spreading from head to tail. These include models for *Xenopus* (*Ferrario et al., 2018*; *Hull et al., 2016*), lamprey (*Kozlov et al., 2009*; *Kozlov et al., 2014*; *Messina et al., 2017*), and salamanders (*Bicanski et al., 2013*; *Ijspeert et al., 2007*). These models have become detailed enough to include many neurons forming circuits distributed across the hindbrain and the spinal cord. Some models incorporate specific intrinsic and ligand-gated currents with known roles in rhythmogenesis in their respective species (*Ferrario et al., 2018*; *Kozlov et al., 2009*; *Kozlov et al., 2014*). Simulations of the models have been used to test aspects of swimming control, including steering commands from descending commands to spinal networks (*Kozlov et al., 2014*), the integration of sensory triggers of locomotion (*Ferrario et al., 2018*; *Ijspeert et al., 2007*), the coupling of axial and limb central pattern generators (*Ijspeert et al., 2007*), and the role of left-right coupling in rhythm generation (*Messina et al., 2017*). Our model could be used to identify possible similarities or differences in how these aspects of motor control are controlled in the zebrafish.

## Testable predictions

To the best of our knowledge, this is the first model to generate several forms of locomotor movements in developing zebrafish based upon previously described neurons and their connectivity patterns. The analysis of the simulations generated yielded several predictions about possible connections between spinal neurons, firing properties of neurons, and roles for neurons in specific locomotor movements. For instance, the single coiling model predicts that IC and V0d are coupled together to facilitate the activation of V0ds, which are responsible for the glycinergic SBs observed in spinal neurons at this stage (*Saint-Amant and Drapeau, 2001*; *Tong and McDearmid, 2012*).

Our modeling study also predicts that the generation of double and even multiple coils depend on untested connections between V2a and V0v neurons and between V0v and IC neurons. The latter connection would be needed to initiate consecutive left-right alternating coils through the activation of contralateral IC neurons, while the former connection would be needed to activate the ipsilateral V0v responsible for the activation of contralateral ICs. The V2a to V0v connection could be deemed unnecessary in light of possible gap junction coupling between ipsilateral IC and V0vs. However, our modeling suggests that delayed activation of V0vs would allow the ipsilateral coil to complete before activating the contralateral coil. This delay would not be possible with gap junction mediated excitation of V0vs by ipsilateral ICs. Our double-coiling model also predicts that contralateral inhibition of ICs by V0ds prevents the generation of multiple coilings. Several of these predictions are supported by pharmacological experiments suggesting that blocking glutamatergic transmission in embryonic zebrafish precludes double coiling while blocking glycinergic transmission at that stage promotes multiple coilings (*Knogler et al., 2014*).

The beat-and-glide model also proposes a prominent role of delayed contralateral excitation in ensuring repetitive left-right alternating tail beats during swimming. Whether delayed contralateral excitation is a conserved mechanism of operation in double coiling and swimming remains to be tested experimentally. While V0v neurons are the likely candidate to mediate the activation of contralateral movements, different subgroups of V0v neurons are probably involved in coiling versus swimming (*Björnfors and El Manira, 2016*; *Jay and McLean, 2019*) considering the two different targets of contralateral excitation involved, namely ICs in coiling and V2as in swimming. The continued presence of left-right tail beats in simulations where the dI6 population of commissural inhibitory neurons was silenced or in simulations with blockade of glycinergic transmission further underscores the need to test the contributions of V0v neurons to left-right alternation.

Finally, the ability of our model to generate beat-and-glide swimming with or without burst firing neurons suggests a possible degeneracy in the operation of spinal swimming circuits of the developing zebrafish. This possibility would be consistent with the well-characterized degeneracy of the nervous system, as reinforced by modeling studies where combinations of intrinsic properties or connectivity can generate the same motor output (*Goldman et al., 2001*; *Taylor et al., 2009*). Many rhythmogenic currents (e.g., NMDA, calcium-dependent potassium currents, and persistent sodium currents) have been implicated in the operation of locomotor circuits of zebrafish (*Song et al., 2020*) and other invertebrate and vertebrate rhythm-generating circuits (*Anderson et al., 2012*;

*Golowasch and Marder, 1992*; *Ryczko et al., 2010*; *Tazerart et al., 2007*; *Zhong et al., 2007*). In addition, while some motor systems rely upon pacemaker neurons, other rhythmic motor systems could also rely on network-based mechanisms (*Del Negro et al., 2010*), further demonstrating the diversity of means by which the nervous system generates rhythmic activity. Whether the spinal circuits for swimming are degenerate or degeneracy is only exhibited in our modeling remains to be tested experimentally. The operation of the spinal swimming circuit in zebrafish may exhibit degeneracy dependent upon specific environmental or physiological conditions (*Vogelstein et al., 2014*) and their resulting neuromodulatory states.

## Future directions

Our models will require integrating additional cell populations and circuitry to capture the full range of locomotor movements of developing zebrafish. The beat-and-glide model only generates swimming within a narrow frequency range. The generation of a broader range of swimming frequency (*McLean and Fetcho, 2009*) will require expanding each cell population into subgroups with different intrinsic properties (*Menelaou and McLean, 2012*; *Song et al., 2018*), rostrocaudal projection patterns, and specific connectivity patterns between subgroups and between cell populations (*Ampatzis et al., 2014*; *Bagnall and McLean, 2014*; *Kimura and Higashijima, 2019*; *Menelaou and McLean, 2019*; *Sengupta et al., 2021*; *Song et al., 2020*). These subgroups, which may arise from different birth dates (*McLean and Fetcho, 2009*; *Satou et al., 2012*), are active at specific swimming frequencies (*McLean et al., 2007*; *McLean et al., 2008*; *McLean and Fetcho, 2009*). There seem to be modules consisting of neurons within each cell population that are active at specific swim frequencies (*Ampatzis et al., 2014*; *Menelaou and McLean, 2019*; *Song et al., 2018*; *Song et al., 2020*). Indeed, previous studies in zebrafish have shown that MNs and V2a neurons are organized in three different modules (linked to slow, medium, and fast MNs) that are differentially recruited according to swim frequency (*Ampatzis et al., 2014*; *Song et al., 2020*). Swim frequency modules likely include commissural excitatory V0v INs (*Björnfors and El Manira, 2016*; *McLean et al., 2008*) and commissural inhibitory INs belonging to either the dI6 or V0d populations (*Satou et al., 2020*). The modeling of additional subgroups, especially in the context of swim-frequency modules, will need to take into account the high specificity of connectivity between subgroups within a cell population (*Menelaou and McLean, 2019*; *Song et al., 2020*) and subgroups belonging to different spinal populations within swim frequency-modules (*Ampatzis et al., 2014*; *Bagnall and McLean, 2014*; *Menelaou and McLean, 2019*; *Song et al., 2020*).

Subgroups within cell populations are not necessarily restricted to those belonging to different swim frequency modules but may also exist between neurons involved in rhythm versus vigor of movement. Subgroups for vigor seem to be present within the V2a (*Menelaou and McLean, 2019*; *Song et al., 2018*) and V0v (*Björnfors and El Manira, 2016*; *Jay and McLean, 2019*; *McLean et al., 2007*) populations. Furthermore, the implementation of circuitry for swimming vigor is likely to necessitate adding the ipsilaterally projecting, inhibitory V2b population (*Callahan et al., 2019*). The circuits for frequency and vigor are likely to interact, as seen by the swimming frequency-dependent action of V1 neurons (*Kimura and Higashijima, 2019*). Frequency and vigor are also likely to be shaped by sensory information. Incorporating spinal neurons that integrate sensory information (*Liu and Hale, 2017*) provided by peripherally located and spinally located sensory neurons (*Böhm et al., 2016*; *Picton et al., 2021*) will provide a more accurate representation of swimming control at the level of the spinal cord.

Finally, the undefined role of specific spinal neuron populations could be studied after being integrated into the model following further characterization. For example, ventral V3 neurons in mouse spinal locomotor networks have been studied using modeling. Those studies suggest an important role for these neurons in left-right coordination in mouse locomotion (*Danner et al., 2019*). Similar computational studies using our model could reveal testable predictions of the role of these neurons (*England et al., 2011*; *Yang et al., 2010*) in zebrafish swimming.

Our models simulate several developmental milestones of the zebrafish locomotor behavior. Iterative changes were made to each model to successively transition from single coiling to double coiling and then to beat-and-glide swimming. This iterative process could be further developed to obtain a higher resolution understanding of the maturation of locomotion in zebrafish. Further transitory models could be built to fill the gaps between our current models (e.g., a model for burst swimming that precedes beat-and-glide swimming). The generation of these additional transitory models

could be coupled with experimental data studying the mechanisms that drive the transition from one milestone to the other (**Brustein and Drapeau, 2005**; **Knogler et al., 2014**; **Lambert et al., 2012**; **Roussel et al., 2020**) to identify specific underlying changes in intrinsic and network properties. Thus, the models presented herein offer invaluable tools to investigate further the mechanisms by which spinal circuits control facets of swimming, including speed, direction, and intensity through interactions within the spinal cord and with supraspinal command centers, as well as the developmental dynamics that ensure proper maturation of movement during development.

# Materials and methods

## Key resources table

| Reagent type (species) or resource | Designation | Source or reference | Identifiers | Additional information |
| --- | --- | --- | --- | --- |
| Software, algorithm | Python | Python | RRID:SCR_008394 | |

## Modeling environment

Modeling was performed using Python 3.6.3 64 bits (RRID:SCR_008394). We did not analyze the early parts of simulations (up to 200 ms) to allow the effects of initial conditions to dissipate.

## Modeling of single neurons

We modeled neurons using a single compartment, simple spiking neuron model developed by **Izhikevich, 2007**. The following general differential equations govern the dynamics of the membrane potential:

$$CV' = k(V - V_r)(V - V_t) - u + I_{syn}$$
$$u' = a(b(V - V_r) - u)$$
$$if\ V = V_{max}, then\ V \leftarrow c, u \leftarrow u + d$$

(1)

Specific active conductances are not included in these models. Instead, values of the parameters $a$, $b$, $c$, $d$, and $V_{max}$ (which respectively represent the time scale of the recovery variable $u$, the sensitivity of $u$ to the subthreshold variation of $V$, the reset value of $V$ after a spike, the reset value of $u$, and the action potential peak), as well as values of the parameters $k$, $C$, $V_r$, and $V_t$ (coefficient for the approximation of the subthreshold part of the fast component of the current-voltage relationship of the neuron, cell capacitance, resting membrane potential, and threshold of action potential firing) can be selected to model a wide range of firing behaviors including bursting (or chattering) pacemaker, tonic firing, phasic spiking neurons, or firing rate adaptation neurons (**Table 4**). $I_{syn}$ represents the sum of the synaptic and gap junction currents received by the neuron. For all models, the Euler method was used for solving ordinary differential equations with a time step of 0.1 ms.

## Modeling synapses

We modeled all electrical synapses (i.e., gap junctions) as ideal resistors following Ohm's Law:

$$I_{gap:pre,post} = V_{pre}G_{pre,post}$$

(2)

With $I_{gap:pre,post}$ representing the synaptic current flowing to the postsynaptic neuron from the presynaptic neuron through gap junctions and $G_{pre,post}$ representing the total conductance of gap junctions between the presynaptic and postsynaptic neurons (**Table 1**).

Synaptic conductances of chemical synapses were modeled as a sum of two exponentials weighted by a synaptic weight based upon the general equation:

$$I_{pre,post} = (V_{post} - E_{rev})\left(e^{-\frac{t-t_0}{\tau_r}} - e^{-\frac{t-t_0}{\tau_f}}\right)W_{pre,post}\ if\ V_{pre} > V_{thr}$$

(3)

where $I_{pre,\ post}$ is the synaptic current received by the postsynaptic neuron from neurotransmitter release by the presynaptic neuron if the presynaptic neuron membrane potential, $V_{pre}$, crosses a

voltage threshold, $V_{thr}$, at the synapse. $V_{post}$ is the membrane potential of the postsynaptic neuron, $E_{rev}$ is the reversal potential, $\tau_r$ and $\tau_f$ are the rise and fall time constants, respectively, $t_0$ is the time at which $V_{pre}$ crossed $V_{thr}$, and $W_{pre, post}$ is the synaptic weight between the presynaptic and postsynaptic neurons (*Table 3*). $I_{pre, post}$ is equal to 0 if $V_{pre}$ is below $V_{thr}$.

We implemented two types of chemical synapses: glutamatergic and glycinergic synapses. The former differs from the latter by the respective reversal potential values $E_{rev}$ of glutamatergic and glycinergic synapses and the time constant values $\tau_r$ and $\tau_f$ (*Table 3*). Values of the glycinergic $E_{rev}$ are depolarized at early developmental stages (*Saint-Amant and Drapeau, 2000*; *Saint-Amant and Drapeau, 2001*), and this reversal potential becomes gradually hyperpolarized (*Ben-Ari, 2002*) as the equilibrium potential of chloride hyperpolarizes in the zebrafish nervous system (*Zhang et al., 2010*). All chemical synapses were turned off in the initial 50 ms of every simulation to allow initial conditions to dissipate.

## Spatial arrangement of spinal neurons

A key feature of our modeling approach was to assign spatial coordinates (*x*, *y*) to point-like neurons (i.e., neurons have no spatial dimension, but they have a position in space), giving the spatial distribution of neurons a central place in our model computing process. We used the Euclidean distance to calculate the distance between each neuron and to approximate axon length. Distance unit is arbitrary and was set so that one model somite was 1.6 arbitrary distance units (a.d.u.) long. Time delays for each synaptic connection were computed as a function of the distance between neurons and were used to calculate delayed synaptic current:

$$I_{delayed:pre,post}(t) = I_{pre,post}\left(t - \frac{D_{pre,post}}{cv}\right) \tag{4}$$

With $D_{pre, post}$ as the Euclidean distance between the presynaptic and postsynaptic neurons and *cv* as the transmission speed in arbitrary distance units per second (a.d.u./s). This distance and the neuron position were also used to apply conditions on synaptic weights of neurons (e.g., limits as to how far descending neurons project). For the single coiling model, *cv* was set to 4.0 a.d.u/s. For the multiple coiling model, *cv* was set to 1.0 a.d.u/s. These values were obtained through trial-and-error and may reflect changes in myelination and body size of the developing zebrafish. For the beat-and-glide swimming model, *cv* was set to 0.8 a.d.u/s, which led to intersegmental transmission delays in the range of 3.0–4.0 ms, closely matching the 1.6 ms intersomitic delay previously reported (*McDearmid and Drapeau, 2006*), assuming that each model somite represents two biological somites at this developmental stage (see *Musculoskeletal model* below).

Spinal locomotor circuits were distributed across two columns, one for each side of the body, giving the network a nearly one-dimensional organization along the rostrocaudal axis. Therefore, we used the *x*-axis as the rostrocaudal axis, whereas the *y*-axis was only used to partition neurons from the left and right sides (assigning the coordinate y=1 a.d.u. for the right side and y=−1 a.d.u. for the left side).

## Sensitivity testing

We scaled key parameters to Gaussian noise to test the robustness of our three base models (single coiling, multiple coiling, and beat-and-glide swimming) to parameter variability. Sensitivity to noise of the base models was tested by scaling the parameters that set the tonic motor command drive's amplitude, the rostrocaudal length of neuron projections, the membrane potential dynamics (Izhikevich model), and synaptic weighting. These four sets of parameters were randomized by multiplying the parameters with a random number picked from a Gaussian distribution with mean=1, and standard deviations, $\sigma_d$, $\sigma_l$, $\sigma_p$, and $\sigma_w$, respectively. The amplitude of the motor command drive was randomized at each time point. The parameters for the membrane potential dynamics, rostrocaudal length of axons, and synaptic weights were randomized at the start of each simulation and did not change during the simulations.

## Musculoskeletal model

We implemented a musculoskeletal model of the fish body to convert the output of the spinal circuit model into changes in body angles and frequency of locomotor movements. Each MN output along

the fish body was inputted into a muscle cell (**Figure 1**). The membrane potential of the muscle (*V*) was modeled as a simple passive RC circuit (*R* and *C* being the muscle cell membrane resistance and capacitance, respectively), described by the following equation:

$$V'_{muscle} = \frac{V}{RC} + \frac{I_{syn}}{C} \tag{5}$$

For muscle cells, values of *R* were 25 (single coiling), 50 (double coiling), and 1 (beat-and-glide); values of *C* were 10 (single coiling), 5 (double coiling), and 3 (beat-and-glide). These values were chosen to produce kinematics representative of those seen experimentally. To reduce computational load, we modeled one muscle cell as representing three somites of the body in the base model for coiling and two somites of the body in the base model for swimming. The whole body of the fish was modeled as a chain of uncoupled damped pendulums. We computed local body angles according to the difference in activity between the local left and right muscle cells. The deflection angle $\theta_i$ of the *i*th muscle cell was computed according to the following differential equation.

$$\theta''_i + 2\zeta\omega_0\theta'_i + \omega_0^2\theta_i = \alpha(1 - 0.2R)(V_{Rmuscle,i} - V_{Lmuscle,i}) \tag{6}$$

With $V_{Rmuscle,i}$ and $V_{Lmuscle,i}$ being the solution of the equation (**Equation 5**) for the *i*th muscle on the right and left side of the body, respectively (**Figure 1D**). $\alpha$ is the conversion coefficient from an electric drive of the muscle cells to a mechanical contraction of the same cells. The midline of the body can be computed at any given time as (*x*, *y*) coordinates using trigonometric identities from $\theta_i$ (**Figure 1E**). Specifically,

$$\begin{aligned} x_i &= x_{i-1} + l \cdot sin(\theta_i) \\ y_i &= y_{i-1} - l \cdot sin(\theta_i) \end{aligned} \tag{7}$$

where $(x_i, y_i)$ are the spatial coordinates of the *i*th somite, and *l* is its length. We set $(x_0, y_0)$ to (0, 0) and applied the previous set of equations (**Equation 7**) for $i \geq 1$. Thus, heat-maps of local body angle ($\theta_i$) variation through time provide comprehensive information about the network output (**Figure 1F**). The integrated motor output of the model (e.g., **Figure 5B**) was calculated as the sum of the muscle output at all muscle cells on both sides of the body, followed by a convolution of this sum with a 50 ms square wave. Left-right alternation at the *i*th somite was analyzed using cross-correlation of $V_{Rmuscle,i}$ and $V_{Lmuscle,i}$ at that somite. The minimum coefficient in the range of time delays between −20 and 20 ms was calculated to estimate left-right alternation. A value of 0 indicates left-right out-of-phase alternation, while a value of 1 suggests complete in-phase synchrony.

## Analysis of locomotor activity

To calculate the duration of swimming episodes, we summated the muscle activity across all somites from both sides of the body. This muscle activity was then convoluted, and a threshold of 0.5 arbitrary units was set to detect the start and end of each swimming episode of most simulations. In a few simulations where motor output was very large, the threshold was adjusted to detect episodes. To estimate the tail beat frequency, we determined when the most caudal somite crossed the midline of the body of the musculoskeletal model (a threshold of 0.5 arbitrary units from the center was used to detect crossing to a side of the body). The reciprocal of the interval between consecutive left-to-right or right-to-left crossing was used to calculate the instantaneous tail beat frequency. Any interval greater than 100 ms was considered to be between episodes rather than within an episode and discarded from the calculation of instantaneous tail beat frequency.

To calculate the phase delay between pairs of neurons in the beat-and-glide swimming model, we first calculated the autocorrelation of the reference neuron. The time delay at which the peak autocorrelation occurred was used to estimate the period of the reference neuron cycle. The cross-correlation between the reference and test neuron was then calculated, and the phase delay was calculated as the time delay at which the peak of the cross-correlation occurred divided by the cycle period of the reference neuron in radians. In the coiling models, the cycle period is 1000–2000 ms (single coiling) or 10,000–20,000 ms (double coiling) due to longer inter-coiling intervals. Normalizing phase shifts by this cycle period makes the phase delays very small. Therefore, for the coiling models, the period of the reference neuron cycle was estimated by the average duration of single

coiling (1074 ms) or double coiling events (1891 ms). Note that this procedure does not change the polarity of the phase delay but better separates the various phase delays on a polar plot.

## Statistical analysis

Statistical analysis was performed using the SciPy Python library. Statistical tests consisted of one-factor ANOVA tests followed by two-tailed Student's $t$-tests. A $p$-value$<0.05$ was used to determine statistical significance, and all tests were corrected for multiple comparisons (Bonferroni correction for multiple $t$-tests).

## Availability of code

The code for the models and for the figures can be accessed on GitHub, copy archived at swh:1:rev: dd36ce928a2775eeed45149444962a422cb16446 (*Roussel et al., 2021*). Updates and revisions to the models will also be made available at this site. The simulation data that was used for the figures can be accessed at the Federated Research Data Repository at the following DOI: https://doi.org/10.20383/102.0498.

## Acknowledgements

The authors would like to thank Martha Bagnall, Vamsi Daliparthi, Joe Fetcho, Sara Goltash, Michael Hildebrand, Alex Laliberte, Nicolas Lalonde, John Lewis, Aaron Shifman, and Emily Standen for advice related to the modeling and critical discussion of this manuscript. This research was supported by an NSERC Discovery Grant (RGPIN-2015-06403), an NSERC Canadian Graduate Scholarship M award (NSERC 712210101627), and a McDonnell Center for Cellular and Molecular Neurobiology Postdoc Fellowship FY21.

## Additional information

### Funding

| Funder | Grant reference number | Author |
| --- | --- | --- |
| Natural Sciences and Engineering Research Council of Canada | RGPIN-2015-06403 | Tuan V Bui |
| McDonnell Center for Systems Neuroscience | FY21 | Mohini Sengupta |
| Natural Sciences and Engineering Research Council of Canada | NSERC 553401-2020 | Stephanie F Gaudreau |

The funders had no role in study design, data collection and interpretation, or the decision to submit the work for publication.

### Author contributions

Yann Roussel, Conceptualization, Resources, Data curation, Software, Formal analysis, Validation, Investigation, Visualization, Methodology, Writing - original draft, Writing - review and editing; Stephanie F Gaudreau, Data curation, Validation, Investigation, Writing - review and editing; Emily R Kacer, Software, Investigation, Writing - review and editing; Mohini Sengupta, Validation, Methodology; Tuan V Bui, Conceptualization, Resources, Data curation, Software, Formal analysis, Supervision, Funding acquisition, Validation, Investigation, Visualization, Methodology, Project administration, Writing - review and editing

### Author ORCIDs

Mohini Sengupta http://orcid.org/0000-0002-5234-8258
Tuan V Bui https://orcid.org/0000-0003-0024-1544

Decision letter and Author response
Decision letter https://doi.org/10.7554/eLife.67453.sa1
Author response https://doi.org/10.7554/eLife.67453.sa2

## Additional files

### Supplementary files

• Transparent reporting form

### Data availability

The code for the models and for the figures, as well as the data used to make the figures, can be accessed at https://github.com/bui-lab/code (copy archived at https://archive.softwareheritage.org/swh:1:rev:3268f1f684937f619b6ad87cd0297f8c7ea66db0). Updates and revisions to the models will also be made available at this site.

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
