## [Decision Letter]

**Acceptance summary:**

As animals mature, their locomotor patterns become varied, more flexible and complex. In this manuscript, Roussel et al., build models of the spinal network in embryonic and larval zebrafish based on experimental data to understand how these animals generate distinct behaviors during development.

**Decision letter after peer review:**

Thank you for submitting your article "Modelling spinal locomotor circuits for movements in developing zebrafish" for consideration by *eLife*. Your article has been reviewed by 3 peer reviewers, one of whom is a member of our Board of Reviewing Editors, and the evaluation has been overseen by Ronald Calabrese as the Senior Editor. The following individual involved in review of your submission has agreed to reveal their identity: Rishikesh Narayanan (Reviewer #2).

*Reviewer #1 (Recommendations for the authors):*

For the B panels in figures 2, 3 and 4, are these responses of isolated neurons or neurons in the fully connected network?

Wherever neuron traces are shown, showing rostrocaudal and left-right neuron traces in the same panel makes them too cluttered. In all cases, where the firing starts and stops is difficult to discern. It may suffice to show one rostral and one caudal neuron trace.

Single coiling:

Include sensitivity analysis in graphical form in Figure 2.

SBs are depolarizing and reverse at -40. How critical are they then for preventing simultaneous contralateral coiling? Why an analysis of neuron deletions has not been done?

Was sensitivity different for glycinergic synapses and electrical synapses?

Was bursting in PMs critical for generating single coiling? This can be included in Figure 2. Figure 2 can be improved to test the single coiling model in multiple ways. In its current form, it doesn't provide much data regarding the performance of the model.

Double coiling:

Line 219: what proportion of PMs were electrically coupled to rostral CENs?

The neuron traces are confusing with too many superimposed.

Line 240: Replace 'with' with 'to'.

What accounts for the long delay in firing between PMs and other neurons though they are electrically coupled? These seem much longer than segmental conduction delays. Such long delays seem critical for CENs to initiate the second coil on the contralateral side once the first coil is over. Yet, this seems inexplicable following known monosynaptic/conduction delays. Likewise, in this circuit configuration, what prevents the circuit from going into third or fourth coils? It is not clear which process decays slowly to disallow such multiple coils. Similarly, it is not clear how under conditions of glycinergic blockade in the model, multiple coilings are generated if inhibition is normally over before CENs are even activated.

Figure 3C-E: Why aren't there multiple traces for IEDs?

Figure 3H: Convert to color for better effect.

Beat and glide swimming:

Line 398: Do you mean 'few'?

Line 431: "Silencing…' – this sentence does not make sense.

How is swimming initiated?

Figure 4A: This schematic is too confusing. Do MNs receive only IED inputs? Do they not receive direct inhibition?

Figure 4C, F: Unable to discern any detail of MN activity due to overlap of traces. Show phase relationships between classes of neurons and comment.

Figure 4D: Same as above, cannot see the color coded information clearly as most angles appear red or blue and therefore are at the extremes of their scalebar. Why aren't there any values for body position 1.0 which should be the tail tip?

Figure 5B: Authors state in line 432 that removing CENs does not affect rhythmic firing in IEDs or MNs. When CENs are removed, is there left-right alternation? From the traces shown, there doesn't appear to be and the video shows what appear to be unilateral tail flicks. I couldn't tell for sure as they are at a fast playback speed. If that is the case, authors need to be more convincing of this result in the figure and the video and perhaps include L-R phase relationships to establish that alternation occurs. What do the peaks in Row3 correspond to? Is each cycle a bout? Seems too slow to be a tail beat? Do the dampened oscillations in the middle epoch qualify as bonafide swimming? If yes, what is the justification? Also please explain, 'sum of left and right motoneuron activity was summed'. Why use this parameter as motor output when you have the muscle transform calculated? Why is MN activity shown on a different scale compared to the other rows below?

Figure 7: The importance of bursting IED is not really clarified in this figure. Is there a quantitative difference in durations? Both long and short are produced. Why? Perhaps a phase plane analysis of contribution of IED parameters to durations will help sort out the underlying dynamics? Young prefeeding larvae do exhibit long duration swims, so the conclusion that bursting IEDs are essential for beat and glide swimming appears too strong for the results presented here.

The videos shown for CIN null and strychnine experiments show left-right tail beats while figure panels show different results.

Discussion:

Lines 604-635: Authors discuss differential recruitment of spinal neurons as a function of 'speed'. However, the studies cited looked at the recruitment of neurons as a function of frequency. It has recently become clear that speed can be changed even without changing swim frequency and that MNs are recruited as a function of forward speed and not frequency (Jha and Thirumalai 2020).

Methods:

A more detailed methods section is warranted.

It is not clear what the authors mean by tonic motor command- is it a DC current injection or a regular synaptic input?

Line 705: What is s here?

Line 739: Why is capacitance decreasing with age?

Line 745: How is deflection angle calculated per somite when three somites are collapsed into one muscle?

Line 756: 'muscle output' is vague – do you mean V or theta?

Figure 1D: The transformation from local body angle to an overall body midline position can be better schematized to help the reader.

Figure 1E: More intuitive to have time on x-axis as has been done in Figure 7F. Change elsewhere also.

*Reviewer #2 (Recommendations for the authors):*

1. The sensitivity analyses involving variability in individual parameters is useful. But, it is not clear from the description (starting line 721) on what specific parameters are governed by \σ_p, \σ_w and \σ_d. Importantly, these sensitivity analyses do not seem to cover cell-to-cell variability in Izhikevich model parameters (spanning a, b, c, d, k, C, V_r, V_t; it is not clear if \σ_p alters any of these parameters?). Specifically, all neurons of the same subtype seem to have the same model parameters. While the firing dynamics of Izhikevich model neuron are critically dependent on the model parameters, there are a range of model parameters (perturbations around the employed values) over which similar firing patterns could be achieved. How sensitive are the conclusions presented here to such variability across neurons of the same subtype in the network (with the variability reflecting electrophysiological recordings from larval neurons)?

What would be the impact of heterogeneities in gap junction connectivity? From the description provided, it seems like the model assumes identical weights for gap junction connectivity. How sensitive would the conclusions presented here to adding variability to gap junctional weights?

Additional sensitivity analyses could focus on the numbers of neurons per chain, number of connections within and across chains and glycinergic synaptic connectivity (there seem to be several unsubstantiated assumptions on these connections; e.g. paragraph spanning lines 126-137).

Apart from these, any parameter that has been fixed to be homogeneous should also be considered in addressing the question on whether the results observed are because of the homogeneous nature of that parameter in the simulations. The authors could consider incorporating some of these simulations involving networks with parametric variability, and mention the others in the discussion.

2. All interpretations of the authors from experimental data are based on the summary statistics, and do not account for heterogeneities across different larvae. For instance, the authors cite experimental data that adding CNQX and APV to block glutamatergic transmission precluded double coils while sparing single coils, and that blocking glycinergic synapses led to triple or even quadruple coils. They use these conclusions to drive their modeling outcomes. However, are the impacts of CNQX or APV or blockade of glycinergic synapses the same across all larvae? Do all larvae behave identically when treated with these pharmacological agents? If not, how do the authors account for these heterogeneities across larvae.

Employing summary statistics to define models or assess outcomes is perilous, and the authors should account for heterogeneities in experimental outcomes to define their models (Marder and Taylor, Nature Neuroscience, 2011). Heterogeneities and variability should be accounted for each measurement at each scale in interpreting modeling and experimental outcomes -- the CNQX/APV/Glycinergic synaptic blockers is just one instance. The ideal way to do this is to generate a population of heterogeneous models and derive conclusions from there (Marder and Taylor, Nature Neuroscience, 2011). I recommend that the authors mention this a future direction, refine their interpretations in the Discussion section accounting for heterogeneities across larvae and present the caveats of using summary statistics to drive experimental/model interpretations.

3. Are there several routes (in terms of neuronal intrinsic properties, neuronal firing patterns, network connectivity, electrical/chemical synapse weights, neuromodulatory tones, etc.) to achieving single coiling or double coiling or other locomotor movements? Or, is there a unique route to achieve these across larvae? Could the authors comment on potential degeneracy across scales in achieving the same behavioral outcomes (see Vogelstein et al., Science, 2014; Edelman and Gally, PNAS, 2001)? How would such degeneracy alter the conclusions presented here? How would such degeneracy relate to heterogeneities across larvae (Pt. 2 above) and variability in parameters (Pt. 1 above)? The authors should provide detailed discussion on these questions. They have considered single hand-tuned models across different developmental stages to make their points and demonstrate behavioral outcomes. But, it is important to ask if this is the only route to achieve these behavioral outcomes, and if zebrafish and their larvae might be using different structural combinations to elicit same functional outcomes?

While the authors have talk about potential ion-channel degeneracy in the discussion, they should also comment on circuit level degeneracy involving different neuronal subtypes and disparate synaptic components (in terms of neuronal intrinsic properties, neuronal firing patterns, network connectivity, electrical/chemical synapse weights, neuromodulatory tones, etc. e.g. Prinz et al., Nat Neuroscience, 2004). The authors could mention in the Discussion section about the possibility of using an unbiased stochastic search involving all parameters to assess such potential degeneracy involving intrinsic properties and synaptic connectivity across all stages of the developing fish. The authors could also mention in the discussion the disadvantages of using a single hand-tuned model and in deriving all interpretations from that single model (Marder and Taylor, Nat Neuroscience., 2011).

4. Please add a separate Discussion section on testable predictions that emerge from the model presented here, which provide clear pointers that would test the overall hypothesis.

*Reviewer #3 (Recommendations for the authors):*

Strengths and weaknesses

The key strength of this manuscript is the detailing of a set of related models detailing the motor output of the larval zebrafish across key stages of development. The models should form a basis for future research. It also a first of its kind – I don't know of similar models focusing on development of locomotor function. The main weakness is the reliance on assumptions of model connectivity. But I suggest that if the model is treated as a basis for the community to refine and validate it will be incredibly useful.

---

## [Author Response]

Reviewer #1 (Recommendations for the authors):For the B panels in figures 2,3 and 4, are these responses of isolated neurons or neurons in the fully connected network?

The responses are from isolated neurons, and this has been clarified in the figure legends of Figures 2-4 and 8.

Wherever neuron traces are shown, showing rostrocaudal and left-right neuron traces in the same panel makes them too cluttered. In all cases, where the firing starts and stops is difficult to discern. It may suffice to show one rostral and one caudal neuron trace.

We have reduced the number of traces in figures depicting neural activity during simulations to a rostral, middle, and caudal neuron.

Single coiling:Include sensitivity analysis in graphical form in Figure 2.

We have added the sensitivity analysis for the single coiling model to Figure 2.

SBs are depolarizing and reverse at -40. How critical are they then for preventing simultaneous contralateral coiling? Why an analysis of neuron deletions has not been done?

Due to the lack of contralateral excitation, there are no contralateral coils in the single coiling model. We further emphasize this point in a new simulation where we remove contralateral inhibition by V0d, which removes SBs, and find no emergence of multiple coilings (Figure 2 —figure supplement 1). We have added an analysis of neuron deletions for ICs and MNs in Figure 2 —figure supplement 1.

Was sensitivity different for glycinergic synapses and electrical synapses?

We did not perform this analysis for the single coiling model as glycinergic synapses do not contribute to the generation of coils or the attenuation of double or multiple coils in the single coiling model (see our response to the previous point). For the double coiling model, we did perform a sensitivity analysis of chemical versus electrical synapses (Figure 3 —figure supplement 2, Lines 356-360).

Was bursting in PMs critical for generating single coiling? This can be included in Figure 2. Figure 2 can be improved to test the single coiling model in multiple ways. In its current form, it doesn't provide much data regarding the performance of the model.

The bursting in the ICs (formerly PMs) is not essential to single coiling. In response to this question, we created a model with ICs firing a single spike per burst. With an increase in the synaptic weight from IC to MNs, this generated single coiling as well. However, we modelled ICs as bursting to resemble the recordings of ICs from Tong and McDearmid (2012), which show sustained bursts of short spikelets. To address this question, we have clarified the reason why our IC models display bursting. We can add the result of the one-spike bursting ICs as a figure supplement to Figure 2 if desired.

Double coiling:Line 219: what proportion of PMs were electrically coupled to rostral CENs?

All ipsilateral ICs (we now refer to PMs as their biological equivalents, the IC neurons) were coupled with the most rostral V0vs (formerly CENs) in the first four rostral somites. We have clarified this point in the text (Line 270).

The neuron traces are confusing with too many superimposed.

As mentioned above, we have reduced the number of traces to a rostral, middle, and caudal neuron.

Line 240: Replace 'with' with 'to'.

This change has been made (Line 268).

What accounts for the long delay in firing between PMs and other neurons though they are electrically coupled? These seem much longer than segmental conduction delays. Such long delays seem critical for CENs to initiate the second coil on the contralateral side once the first coil is over. Yet, this seems inexplicable following known monosynaptic/conduction delays. Likewise, in this circuit configuration, what prevents the circuit from going into third or fourth coils? It is not clear which process decays slowly to disallow such multiple coils. Similarly, it is not clear how under conditions of glycinergic blockade in the model, multiple coilings are generated if inhibition is normally over before CENs are even activated.

For double coiling, the critical delay is the delay between IC firing and the subsequent activation of the contralaterally-projecting glutamatergic V0v (formerly CENs). This delay is a sum of the delay in ICs activating V2a (formerly IEDs) and the latter activating V0v through glutamatergic excitation. The V2a to V0v excitation has to be titrated carefully to produce a delay. Otherwise, there is no second contralateral coil if the first coil is not allowed to be completed. To highlight this delay, we have added a phase delay plot (Figure 3D) and also used simulations where the V2a to V0v is increased (Figure 3E) to support these points further (Lines 305-312).

Figure 3C-E: Why aren't there multiple traces for IEDs?

In our model, the more caudal V2as (formerly IEDs) are not activated. This lack of activity in the caudal V2as does not limit the ability to generate double coils. The rostral V0vs are crucial for this behaviour and are recruited by rostral V2as.

Figure 3H: Convert to color for better effect.

This change has been made (now Figure 3I), and we use this color scheme for new figures depicting additional sensitivity testing with the double coiling model (Figure 3J-M and Figure 3 —figure supplement 2).

Beat and glide swimming:Line 398: Do you mean 'few'?

No, by ‘new’, we meant that this connection from V2as and V1s was new. However, since V1s were new, any connections to and from them are new. To avoid any confusion, we removed “new”. “Reciprocally, V2as formed glutamatergic synapses to caudally located V1.” (Line 429).

Line 431: "Silencing…' – this sentence does not make sense.

This sentence has been modified to better convey our message:

“Silencing V0vs diminished but did not eliminate the rhythmic firing of V2as or MNs. During Epoch 2, the tonic motor command continued to activate V2as. Pairs of left-right tail beats may result from the commissural inhibition by dI6s that was still present. However, removing the contralateral excitation by V0v prevented the repetitive activation of the silent side after each tail beat, which severely reduced episode duration and the number of tail beats generated in each episode (Figure 5G-L, Figure 5 —figure supplement 1G-L, Figure 5 – video 1).” (Lines 505-510).

How is swimming initiated?

The new phase delay plot that we have added suggests that swimming is initiated by V2as that serve to drive all other spinal neurons in each tail beat (Figure 4H and Lines 486-489).

Figure 4A: This schematic is too confusing. Do MNs receive only IED inputs? Do they not receive direct inhibition?

The schematic for the beat and glide model has been modified. The inhibition of MNs by V1s (formerly IIA) has been added for clarification.

Figure 4C, F: Unable to discern any detail of MN activity due to overlap of traces. Show phase relationships between classes of neurons and comment.

The phase relationships have been added in Figure 4H, and these results are commented on (Lines 486-496).

Figure 4D: Same as above, cannot see the color coded information clearly as most angles appear red or blue and therefore are at the extremes of their scalebar. Why aren't there any values for body position 1.0 which should be the tail tip?

We agree that the colour-coding of the angles does not allow for a precise depiction of the angles. Because of the time range depicted, we use this type of heatmap (also used in Figure 8 and Figure 8 —figure supplement 1) mainly to show which somites are active, depict the swimming pattern and demonstrate that episodes consist of left-right tail beats. The more precise changes in angles across the somites are better viewed in the videos, which have been slowed down to observe the kinematics better.

The omission of the values for body position at the tail tip for the local body angle heat maps has been corrected, and they are now present.

Figure 5B: Authors state in line 432 that removing CENs does not affect rhythmic firing in IEDs or MNs. When CENs are removed, is there left-right alternation? From the traces shown, there doesn't appear to be and the video shows what appear to be unilateral tail flicks. I couldn't tell for sure as they are at a fast playback speed. If that is the case, authors need to be more convincing of this result in the figure and the video and perhaps include L-R phase relationships to establish that alternation occurs. What do the peaks in Row3 correspond to? Is each cycle a bout? Seems too slow to be a tail beat? Do the dampened oscillations in the middle epoch qualify as bonafide swimming? If yes, what is the justification? Also please explain, 'sum of left and right motoneuron activity was summed'. Why use this parameter as motor output when you have the muscle transform calculated?

We have revised the text to clarify that there is no left-right alternation without commissural excitation. Furthermore, we have added a supplementary figure with membrane potential traces of all neurons to clarify that while there is still rhythmic activity in V2a and MNs, this activity is greatly diminished, and swimming episodes are considerably shortened. We have added supplementary figures with membrane potential traces of all neurons for all the knockout simulations (Figure 5 —figure supplement 1, and Figure 6 —figure supplement 2).

We have corrected the statement in the original version of the manuscript regarding using the sum of left and right motoneuron activity. It is the sum of left and right muscle output.

Why is MN activity shown on a different scale compared to the other rows below?

We show MN activity on a different scale to show a magnified view of MN activity during one or two swim episodes. In the rows below, the motor output and parameters of swimming activity are displayed for the entire simulation. This change in time scale is to show fluctuations of this output during the progression of the simulations.

Figure 7: The importance of bursting IED is not really clarified in this figure. Is there a quantitative difference in durations? Both long and short are produced. Why? Perhaps a phase plane analysis of contribution of IED parameters to durations will help sort out the underlying dynamics? Young prefeeding larvae do exhibit long duration swims, so the conclusion that bursting IEDs are essential for beat and glide swimming appears too strong for the results presented here.

As described in the introduction to our Author’s Response, the modified beat-and-glide swimming model does not suggest that bursting IEDs are essential anymore, so this section has been removed.

The videos shown for CIN null and strychnine experiments show left-right tail beats while figure panels show different results.

While the figures show reduced left-right alternation in the dI6 (formerly CIN null) and strychnine simulations, the videos do indeed still show left-right tail beats (Figure 6 – video 2 and Figure 7 – video 1). The conserved left-right alternation in these simulations is explained by the presence of some left-right alternation in rostral segments. With the left-right alternation in rostral segments, the rest of the body will inevitably sway from left to right. We have added the left-right cross-correlation for a rostral somite to show that silencing dI6 and strychnine have a lesser effect on rostral segments, which drives the continued presence of left-right tail beats. We have also added two figure supplements, Figure 6 —figure supplement 2 and Figure 7 —figure supplement 1, to illustrate that the kinematics of the tail beats are modified in the dI6 null and the strychnine simulations by the reduced, but not eliminated, left-right alternation.

To the best of our knowledge, the experimental work with dI6 silencing (Satou et al., 2020) showed similar levels of reduced left-right alternation as we do. Their results were obtained from ex-vivo recordings. They do not report the effect on actual swimming, so we do not know whether their transgenic fish still show some left-right tail beats. As for strychnine, the effects of strychnine on actual fish swimming are complicated by its effects on neuromuscular junction transmission (E.g. García-Colunga and Miledi, PNAS, 1999,https://doi.org/10.1073/pnas.96.7.4113).

Discussion:Lines 604-635: Authors discuss differential recruitment of spinal neurons as a function of 'speed'. However, the studies cited looked at the recruitment of neurons as a function of frequency. It has recently become clear that speed can be changed even without changing swim frequency and that MNs are recruited as a function of forward speed and not frequency (Jha and Thirumalai 2020).

Thank you for clarifying this important distinction. We now refer specifically to swimming frequency rather than swimming speed. We have modified the text to reflect this distinction.

Methods:A more detailed methods section is warranted.

We have made the methods more detailed. In particular, we have added details about how the muscle activity is converted to body angles (Lines 934 – 950) and how swimming activity was analyzed (e.g. phase delays, episode durations, inter-episode intervals, etc) at Lines 952-974.

It is not clear what the authors mean by tonic motor command- is it a DC current injection or a regular synaptic input?

We modelled the tonic motor command as a DC current injection. This is clarified at Lines 456-458 While this is less biologically accurate, we made this choice to reduce computational load.

Line 705: What is s here?

The s stands for second, and we have clarified this:

“…the transmission speed in arbitrary distance units per second (a.d.u./s)” (Line 883).

Line 739: Why is capacitance decreasing with age?

These differences in capacitance values are a weakness of our simple musculoskeletal model. We found that the lower capacitance values enabled shorter time constants for the high-frequency tail beats instead of the slower coiling movements' speeds. The lower capacitance may reflect differences in muscle types active during coiling versus swimming or kinetics of neuromuscular transmission at stages of development.

Line 745: How is deflection angle calculated per somite when three somites are collapsed into one muscle?

We have clarified the notion of model muscle cell and biological somite so that the deflection angle is calculated for one muscle cell in which are collapsed three somites (coiling models) or two somites (swimming model) (Lines 125-127, 398-400, 922-924).

Line 756: 'muscle output' is vague – do you mean V or theta?

This issue was clarified to mean V:

“Left-right alternation at a particular somite was analyzed using cross-correlation of VRmuscle,i and VLmuscle,ith left and right muscle output at that somite.” (Lines 946-947).

Figure 1D: The transformation from local body angle to an overall body midline position can be better schematized to help the reader.

We have modified this figure to make the transformation more intuitive.

Figure 1E: More intuitive to have time on x-axis as has been done in Figure 7F. Change elsewhere also.

Time has been placed on the x-axis as suggested for this and other similar figures (Figure 8 and Figure 8 —figure supplement 1).

Reviewer #2 (Recommendations for the authors):1. The sensitivity analyses involving variability in individual parameters is useful. But, it is not clear from the description (starting line 721) on what specific parameters are governed by \σ_p, \σ_w and \σ_d. Importantly, these sensitivity analyses do not seem to cover cell-to-cell variability in Izhikevich model parameters (spanning a, b, c, d, k, C, V_r, V_t; it is not clear if \σ_p alters any of these parameters?). Specifically, all neurons of the same subtype seem to have the same model parameters. While the firing dynamics of Izhikevich model neuron are critically dependent on the model parameters, there are a range of model parameters (perturbations around the employed values) over which similar firing patterns could be achieved. How sensitive are the conclusions presented here to such variability across neurons of the same subtype in the network (with the variability reflecting electrophysiological recordings from larval neurons)?What would be the impact of heterogeneities in gap junction connectivity? From the description provided, it seems like the model assumes identical weights for gap junction connectivity. How sensitive would the conclusions presented here to adding variability to gap junctional weights?Additional sensitivity analyses could focus on the numbers of neurons per chain, number of connections within and across chains and glycinergic synaptic connectivity (there seem to be several unsubstantiated assumptions on these connections; e.g. paragraph spanning lines 126-137).Apart from these, any parameter that has been fixed to be homogeneous should also be considered in addressing the question on whether the results observed are because of the homogeneous nature of that parameter in the simulations. The authors could consider incorporating some of these simulations involving networks with parametric variability, and mention the others in the discussion.

As suggested by the reviewer, we have added several sensitivity analyses. These analyses include sensitivity to variability in just the chemical synapses as well as just the gap-junction weights (double coiling model), the rostrocaudal extent of axon projections (all models), and glycinergic reversal potential (double coiling and beat-and-glide swimming models). We have also tested our ability to generate single and double coiling with a thirty-somite model and beat-and-glide with a thirty-somite model and a shorter 10-somite model.

We have also clarified how the sensitivity analysis was performed to test variability of the parameters setting membrane potential dynamics (σp) "…the parameters that set the dynamics of the membrane potential of each neuron (a, b, c, d, and Vmax, k, C, Vr and Vt , see Materials and methods)" (Lines 204-207).

With regards to homogeneity of the parameters in our model, it is true that in the base model, the parameters are homogeneous for each cell within a population or each specific synaptic connection. In our sensitivity analysis, however, every parameter (whether setting membrane potential dynamics, synaptic weight) is scaled individually. We try to describe this “Variability in the amplitude of the tonic motor command, the rostrocaudal extent of every axonal projection, every parameter that set the dynamics of the membrane potential of each neuron (a, b, c, d, and Vmax, k, C, Vr and Vt), and all of the weights of gap junction and chemical synapses were modelled by scaling each value by a random number picked for each simulation” (Lines 204-208).

We now discuss the homogeneity of some of the parameters in our model and the implications of this homogeneity and other sensitivity analyses to be considered in the future (Lines 676-689).

2. All interpretations of the authors from experimental data are based on the summary statistics, and do not account for heterogeneities across different larvae. For instance, the authors cite experimental data that adding CNQX and APV to block glutamatergic transmission precluded double coils while sparing single coils, and that blocking glycinergic synapses led to triple or even quadruple coils. They use these conclusions to drive their modeling outcomes. However, are the impacts of CNQX or APV or blockade of glycinergic synapses the same across all larvae? Do all larvae behave identically when treated with these pharmacological agents? If not, how do the authors account for these heterogeneities across larvae.Employing summary statistics to define models or assess outcomes is perilous, and the authors should account for heterogeneities in experimental outcomes to define their models (Marder and Taylor, Nature Neuroscience, 2011). Heterogeneities and variability should be accounted for each measurement at each scale in interpreting modeling and experimental outcomes -- the CNQX/APV/Glycinergic synaptic blockers is just one instance. The ideal way to do this is to generate a population of heterogeneous models and derive conclusions from there (Marder and Taylor, Nature Neuroscience, 2011). I recommend that the authors mention this a future direction, refine their interpretations in the Discussion section accounting for heterogeneities across larvae and present the caveats of using summary statistics to drive experimental/model interpretations.

Thank you for raising the importance of accounting for inter-animal variability in our modelling work. This issue is now discussed (Lines 683-689).

3. Are there several routes (in terms of neuronal intrinsic properties, neuronal firing patterns, network connectivity, electrical/chemical synapse weights, neuromodulatory tones, etc.) to achieving single coiling or double coiling or other locomotor movements? Or, is there a unique route to achieve these across larvae? Could the authors comment on potential degeneracy across scales in achieving the same behavioral outcomes (see Vogelstein et al., Science, 2014; Edelman and Gally, PNAS, 2001)? How would such degeneracy alter the conclusions presented here? How would such degeneracy relate to heterogeneities across larvae (Pt. 2 above) and variability in parameters (Pt. 1 above)? The authors should provide detailed discussion on these questions. They have considered single hand-tuned models across different developmental stages to make their points and demonstrate behavioral outcomes. But, it is important to ask if this is the only route to achieve these behavioral outcomes, and if zebrafish and their larvae might be using different structural combinations to elicit same functional outcomes?While the authors have talk about potential ion-channel degeneracy in the discussion, they should also comment on circuit level degeneracy involving different neuronal subtypes and disparate synaptic components (in terms of neuronal intrinsic properties, neuronal firing patterns, network connectivity, electrical/chemical synapse weights, neuromodulatory tones, etc. e.g. Prinz et al., Nat Neuroscience, 2004). The authors could mention in the Discussion section about the possibility of using an unbiased stochastic search involving all parameters to assess such potential degeneracy involving intrinsic properties and synaptic connectivity across all stages of the developing fish. The authors could also mention in the discussion the disadvantages of using a single hand-tuned model and in deriving all interpretations from that single model (Marder and Taylor, Nat Neuroscience., 2011).

We agree that there is likely a substantial amount of degeneracy in networks capable of achieving the motor output desired. We show some of this degeneracy by showing that beat and glide can be generated by models with bursting V2a (formerly IED) or V0v (formerly CENs), as well as just tonic neurons. We now discuss this at greater length, in part in light of inter-animal heterogeneity and variability (Lines 742-757).

4. Please add a separate Discussion section on testable predictions that emerge from the model presented here, which provide clear pointers that would test the overall hypothesis.

As suggested by the reviewer, a section on testable predictions has been added to the Discussion (Lines 706-757).